# Mammalian mitohormesis: from mitochondrial stressors to organismal benefits

Amanda L Gunawan [1,2], Irene Liparulo [1,2✉] & Andreas Stahl [1✉]

## Abstract

A variety of stressors, including environmental insults, pathological conditions, and transition states, constantly challenge cells that, in turn, activate adaptive responses to maintain homeostasis. Mitochondria have pivotal roles in orchestrating these responses that influence not only cellular energy production but also broader physiological processes. Mitochondria contribute to stress adaptation through mechanisms including induction of the mitochondrial unfolded protein response (UPR^mt) and the integrated stress response (ISR). These responses are essential for managing mitochondrial proteostasis and restoring cellular function, with each being tailored to specific stressors and cellular milieus. While excessive stress can lead to maladaptive responses, mitohormesis refers to the beneficial effects of low-level mitochondrial stress. Initially studied in invertebrates and cell cultures, recent research has expanded to mammalian models of mitohormesis. In this literature review, we describe the current landscape of mammalian mitohormesis research and identify mechanistic patterns that result in local, systemic, or interorgan mitohormesis. These investigations reveal the potential for targeting mitohormesis for therapeutic benefit and can transform the treatment of diseases commonly associated with mitochondrial stress in humans.

**Keywords** Mitohormesis; Integrated Stress Response; Mitochondrial Unfolded Protein Response (UPR^mt); Mitochondrial Retrograde Signaling; Mammalian Models
**Subject Categories** Metabolism; Translation & Protein Quality

## Introduction

Cells are constantly challenged by pathological, environmental, and transitional stresses, and in response to these stressors, they may employ different strategies to restore cellular homeostasis. Mitochondria, with their multifaceted role, emerge as key organelles in cellular stress adaptation. In fact, the exponential growth of exciting discoveries and tools over the last three decades has significantly advanced the classic and simplistic paradigm of mitochondria as the "powerhouse of the cell" (Fig. 1). Beyond their canonical role in energy production, mitochondria are hubs for sensing cellular stress and initiating adaptive responses, exhibiting diverse functional and molecular features (mitotypes) within tissues, cell types, and subcellular levels (Collins et al, 2002; Han et al, 2023; Rausser et al, 2021). Interestingly, mitochondria have been found to specialize into distinct subpopulations to meet and sustain the specific bioenergetic demands of individual cells (Benador et al, 2018; Ryu et al, 2024).

While the concept of the mitochondrial information processing system (MIPS) has been recently introduced (Picard and Shirihai, 2022), mitochondria have long been recognized for their role in intercellular communication and participation in cell fate decisions. The reduction of mitochondrial membrane potential and release of cytochrome c from mitochondria was identified almost three decades ago as a key signal triggering a cascade of pathways that culminate in apoptosis (Jiang et al, 1999; Liu et al, 1996; Zamzami et al, 1995). Furthermore, additional mitochondrial signaling mechanisms have since been discovered. Mitochondria can sense a wide range of environmental and endogenous inputs, resulting in changes to mitochondrial bioenergetics and morphology. This ultimately produces signals or outputs that can affect other organelles and cause systemic regulation (Picard and Shirihai, 2022) (Fig. 2A). Intriguingly, these signals are not always confined to local or neighboring organelles or cells but can also influence distal organs with systemic outcomes. This confirms that mitochondrial adaptations extend far beyond individual cells: mitochondrial signaling contributes to cellular survival under challenging conditions and influences broader physiological processes (Bar-Ziv et al, 2020).

Among the diverse mechanisms that cells employ to cope with internal and external challenges, several evolutionarily conserved stress response pathways have been characterized. These include the Heat Shock Response (HSR) (Richter et al, 2010), the Unfolded Protein Response of the Endoplasmic Reticulum (UPR^ER) (Harding et al, 1999), the Integrated Stress Response (ISR), and the mitochondrial Unfolded Protein Response (UPR^mt). Each of these pathways responds to distinct types of cellular dysfunction but often converges on shared signaling hubs to determine cell fate. Notably, in mammals the ISR and UPR^mt are activated concomitantly by changes in response to mitochondrial dysfunction, including mitochondrial proteostasis, membrane potential, and metabolic status. Together, these pathways, which we will explore in detail in this review, orchestrate cellular reprogramming aimed at restoring mitochondrial and cellular homeostasis.

[1]Department of Nutritional Sciences and Toxicology, University of California, Berkeley, CA 94720, USA. [2]These authors contributed equally: Amanda L Gunawan, Irene Liparulo.
✉E-mail: irene.liparulo@berkeley.edu; astahl@berkeley.edu

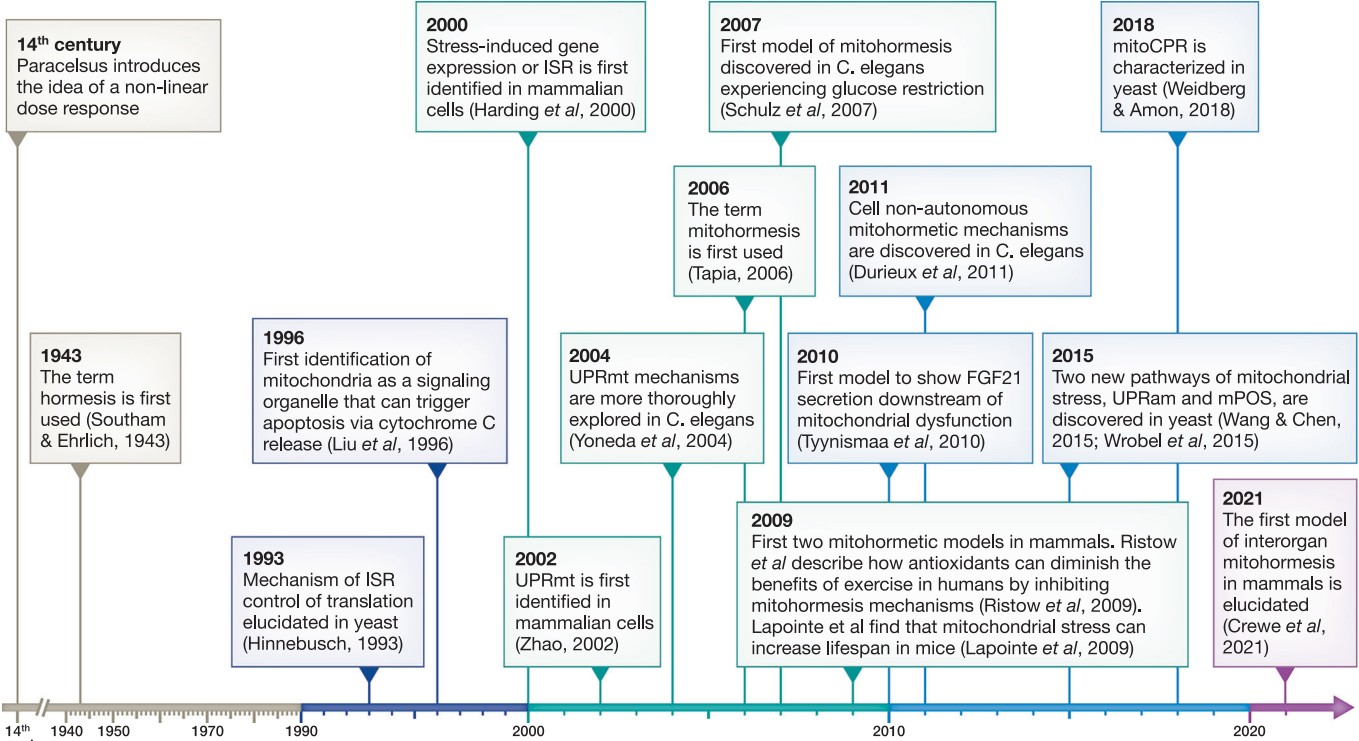

**Figure 1. Timeline of mitohormesis milestones.**

(1) 14th century: Paracelsus introduces the idea of a nonlinear dose response. (2) 1943: The term hormesis is first used (Southam and Ehrlich, 1943). (3) 1993: Mechanism of ISR control of translation elucidated in yeast (Hinnebusch, 1993). (4) 1996: First identification of mitochondria as a signaling organelle that can trigger apoptosis via cytochrome C release (Liu et al, 1996). (5) 2000: Stress-induced gene expression or ISR is first identified in mammalian cells (Harding et al, 2000). (6) 2002: UPRmt is first identified in mammalian cells (Zhao, 2002). (7) 2004: UPRmt mechanisms are more thoroughly explored in *C. elegans* (Yoneda et al, 2004). (8) 2006: The term mitohormesis is first used (Tapia, 2006). (9) 2007: First model of mitohormesis discovered in *C. elegans* experiencing glucose restriction (Schulz et al, 2007). (10) 2009: First two mitohormetic models in mammals. Ristow et al describe how antioxidants can diminish the benefits of exercise in humans by inhibiting mitohormesis mechanisms (Ristow et al, 2009). Lapointe et al find that mitochondrial stress can increase lifespan in mice (Lapointe et al, 2009). (11) 2010: First model to show FGF21 secretion downstream of mitochondrial dysfunction (Tyynismaa et al, 2010). (12) 2011: Cell non-autonomous mitohormetic mechanisms are discovered in *C. elegans* (Durieux et al, 2011). (13) 2015: Two new pathways of mitochondrial stress, UPRam and mPOS, are discovered in yeast (Wang and Chen, 2015; Wrobel et al, 2015). (14) 2018: mitoCPR is characterized in yeast (Weidberg and Amon, 2018). (15) 2021: The first model of interorgan mitohormesis in mammals is elucidated (Crewe et al, 2021).

Given their central role in stress sensing and adaptation, mitochondria are not only targets but also active regulators of these signaling pathways. These stress response strategies highlight orchestrated mechanisms that temporarily halt energetically costly cellular functions while simultaneously enhancing essential pro-survival processes. However, not every stressor exerts the same consequence. Indeed, the magnitude and severity of the stress, in terms of time and intensity, play a role in determining the cell's fate. When stress becomes overwhelming, and the energy required to repair the damage caused by insults is too high or offers little benefit, cells may opt for other strategies. In fact, restoring the cell's status quo may involve clearing damaged organelles, such as selective autophagy of mitochondria (mitophagy), or initiating cell death mechanisms like apoptosis. Moreover, it has been reported that the metabolic state and nature of the mitochondrial defect could lead to ISR activation, revealing that stress response coordination is tailored to specific circumstances (Mick et al, 2020).

There is growing evidence that triggering adaptive responses to mitochondrial stress can confer metabolic advantages (Yi et al, 2018) and extend lifespan. As early as the 14th century, the physician and chemist Paracelsus introduced the idea of a nonlinear dose response, stating "solely the dose determines that a thing is not a poison" (Grandjean, 2016). Hormesis, a term rooted in the Greek word ὁρμᾶν (hormán), meaning "to set in motion" or "to excite," describes a unique concept in toxicology. It refers to a non-traditional dose–response curve in which lower doses of a stressor or toxin can actually provide benefits to an organism, while higher doses tend to be harmful (Calabrese, 2004; Calabrese and Baldwin, 2003). This concept extends far beyond toxicology and is applied to mitochondrial stress, where it is known as mitohormesis (Tapia, 2006). Mitohormesis describes a process where low levels of mitochondrial stress can trigger a mitochondria-mediated cellular response, enabling the organism to recover or compensate from an insult to the mitochondria. This response can result in organismal benefits or protection from future insults. However, higher doses of mitochondrial stress can lead to a maladaptive response, causing cell death, organ dysfunction, pathology progression and onset and general adverse outcomes to organismal health (Fig. 2B). Although the dose–response concept is not novel and it is applicable to various biological phenomena, its application within the field of mitohormesis poses unique challenges and offers many facets for exploration.

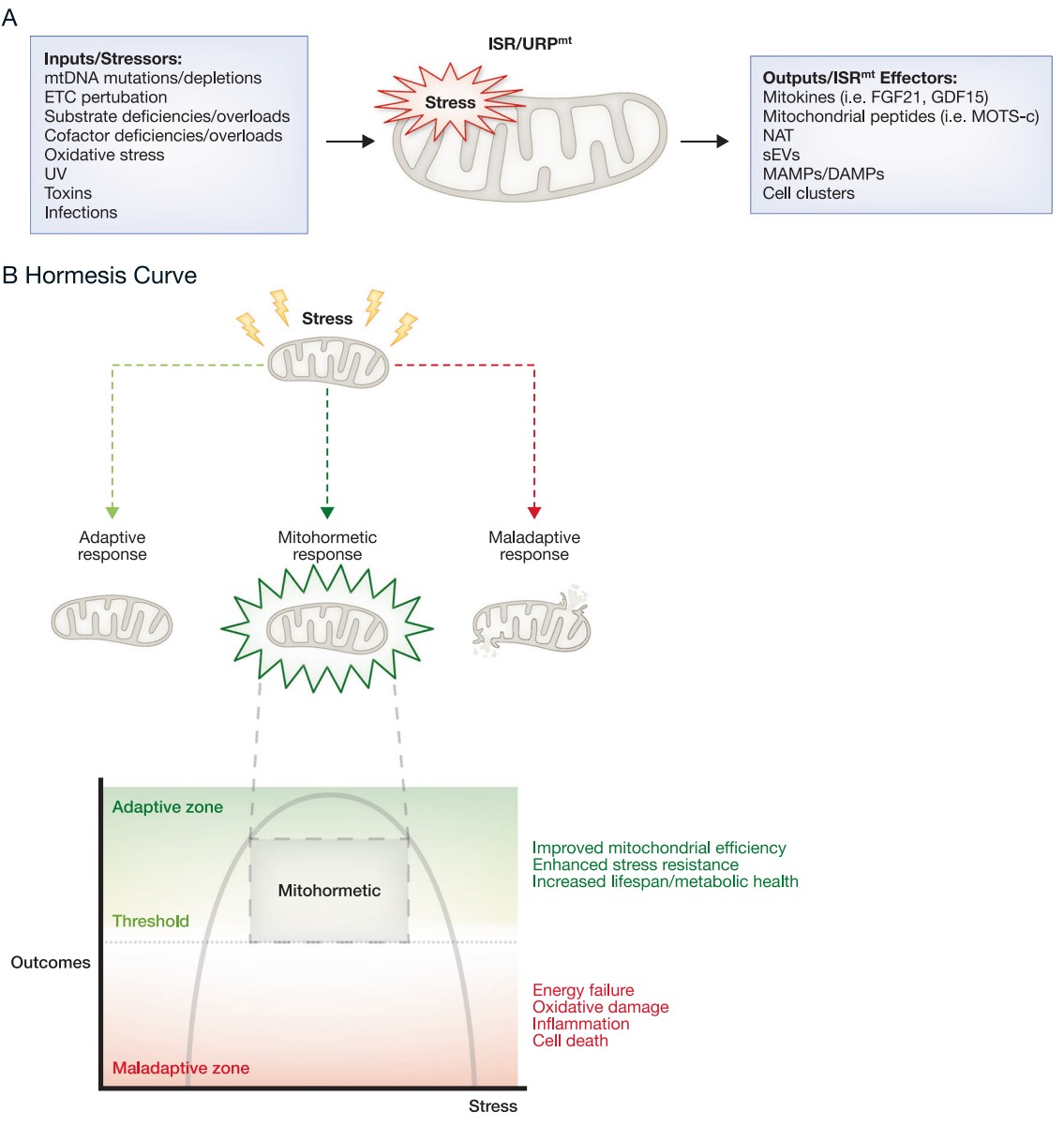

**Figure 2.  Mitochondrial stress pathways in mammalian cells.**

(A) In mammalian cells, both the integrated stress response (ISR) and mitochondrial unfolded protein response (UPRmt) are upregulated in response to a variety of stressors/inputs. The coordinated upregulation of these parallel pathways leads to the secretion of effector/output signals that can affect both local and systemic metabolism, leading to mitohormesis. (B) The nonlinear dose–response curve is a feature of mitohormesis. There exists an optimal dose of stress that results in beneficial mitohormetic phenotypes such as improved mitochondrial efficiency, enhanced stress resistance, and increased healthspan, highlighted as the mitohormetic zone in the shaded box. Too little stress is often not enough to trigger activation of the stress response, while too much stress cannot be overcome/compensated for, resulting in maladaptive responses such as energy failure, oxidative damage, inflammation, and cell death. Timing also plays a role. Transient stressors can activate the stress response and be resolved, while chronic stressors are ultimately detrimental to cells.

In order to understand how mitochondrial stress responses can ultimately result in beneficial results for organisms, further characterization of molecular mechanisms and translation of in vitro research to in vivo models is needed. Thus, the integration of these approaches could dramatically accelerate our current knowledge of ISR. Research on the mitochondrial stress response in organisms such as *C. elegans* and yeast have established pivotal foundations of many mechanisms that are translatable and conserved in more complex organisms. Despite growing interest

in the UPRmt and ISR, many questions about their roles in human health and disease remain unanswered. To bridge the gap between findings in simpler organisms and their relevance in more complex systems, it is crucial to expand research efforts into mammalian models. Such research could clarify the mechanisms behind mitohormesis and guide the development of therapeutic strategies aimed at addressing mitochondrial stress responses. This review offers an updated overview of in vivo mitohormesis models in mammals. We illustrate how mitochondria respond to various

stressors, thereby impacting physiological processes and inducing local and systemic mitohormetic effects. To conclude our perspective, we discussed the clinical implications and highlighted key questions in the field that remain unanswered. While this review focuses specifically on mitohormesis in vivo in mammalian models, we will not discuss the characterization of mitohormetic studies conducted in non-mammalian models, those focused solely on in vitro systems, or research on mitochondrial stress signals that lead to detrimental outcomes for the organism, due to space limitations. Instead, we recommend consulting relevant and notable reviews on these specific topics (Anderson and Haynes, 2020; Bar-Ziv et al, 2020; Cheng et al, 2023; Monzel et al, 2023).

## Coordinated stress responses: understanding ISR and UPR^mt mechanisms in mitochondrial adaptation

While cellular stress responses share the common goal of adapting to or restoring the cells' status quo, they are diverse and tailored to specific challenges. The two stress response pathways that are integral to the mitochondrial stress response and mitohormesis in mammals are the ISR and the UPR^mt (Fig. 3). The activation of these stress responses often reflects a tuned balance between adaptation and dysfunction. While transient or moderate activation can lead to mitohormesis, promoting cellular resilience and restoring homeostasis, excessive or prolonged activation may shift this balance toward maladaptation. The magnitude, duration, and context of the stress response are critical determinants of its outcome. When chronically upregulated, these pathways can drive persistent metabolic reprogramming, mitochondrial impairment, and cellular decline, ultimately contributing to disease onset and progression (Fig. 2B). Understanding the balance between these two outcomes and the different arms of the ISR and UPR^mt that contribute to these two varying results, is essential if we want to harness mitohormesis for therapeutic benefit.

Albeit the ISR and UPR^mt are distinct pathways, they are functionally interconnected. Unlike in yeast and *C. elegans*, where the UPR^mt is mainly responsible for stress resolution following mitochondrial dysfunction, in mammals, the UPR^mt is thought to always be upregulated in concert with the ISR (Fiorese et al, 2016). Previous studies show that the ISR plays a more important role in mammals than in *C. elegans* (Anderson and Haynes, 2020) in response to mitochondrial dysfunction. Mitochondrial stress can activate the ISR, leading to ATF4-dependent transcriptional changes that contribute to UPR^mt activation (Neill and Masson, 2023). However, the UPR^mt also involves ISR-independent mechanisms specific to mitochondrial quality control. These two stress responses are key to the following mammalian mitohormesis studies discussed in this review.

In the early 2000s, Ron and Harding (Harding et al, 2000) coined the term ISR for the first time and provided a comparison between mammalian stress-responsive pathways and yeast general control pathways, highlighting the evolutionary conservation of the organization of these adaptive responses. The ISR is a general stress response pathway that is activated by diverse stressors, including amino acid deprivation, endoplasmic reticulum (ER) stress, mitochondrial stress, and viral infection. Different stress stimuli are sensed by four specialized serine-threonine kinases: Eukaryotic

translation initiation factor 2-alpha kinase 3 (PERK), Protein Kinase R (PKR), General Control Nonderepressible 2 (GCN2), and heme-regulated inhibitor (HRI). In mammals, mitochondrial stress activates different stress kinases depending on the specific mitochondrial stressor. The mechanisms by which each stress kinase is activated by mitochondrial stress are visualized in Fig. 3. One key mechanism that has been recently characterized is how mitochondrial stress can activate HRI via the OMA1–DELE1 axis (Fessler et al, 2020; Guo et al, 2020). Mitochondrial dysfunction can be sensed by the OMA1 Zinc Metallopeptidase (OMA1), which sits on the inner mitochondrial membrane. Once active, OMA1, which is located on the mitochondrial inner membrane, cleaves DAP3-binding cell death enhancer 1 (DELE1) from its long form to a short form. Short form DELE1 (DELEs) then accumulates in the cytosol, where it binds to HRI, causing its activation and phosphorylation, resulting in the initiation of the ISR (Fessler et al, 2020; Guo et al, 2020). Further study of the DELE1-HRI-ISR pathway has also revealed specific conditions, such as iron deficiency, whereby OMA1 cleavage is unnecessary for DELE1-mediated activation of HRI (Fessler et al, 2022; Sekine et al, 2023). In this alternative pathway, there is an import arrest of DELE1 into the mitochondrial matrix, causing it to associate with the mitochondria without being imported. The full-length form of DELE1 can then directly bind to and activate HRI in the cytosol. Additional research needs to be conducted to identify specific stressors that trigger DELE1 cleavage versus a DELE1 import block to activate HRI.

PKR has also been shown to be activated upon mitochondrial stress. Previously, PKR had been known to be activated by exogenous viral double-stranded RNAs, but a recent study found that endogenous RNAs could also activate PKR. Specifically, during mitochondrial stress, mitochondrial RNAs (mtRNAs) have been shown to leak or be transported out into the cytosol, where they can activate PKR (Sud et al, 2016). GCN2 and PERK have also been implicated in the activation of the ISR during mitochondrial stress, although the mechanisms through which this occurs have yet to be elucidated. However, authors have noted that ROS production is essential for activation of these stress kinases under mitochondrial stress conditions (Samluk et al, 2019; Wang et al, 2016). Once these stress kinases are activated, they phosphorylate the eukaryotic translation initiation factor 2 subunit alpha (eIF2α). This crucial phosphorylation event not only decreases protein production but also triggers the upregulation of key transcription factors, such as ATF4, which are essential for the ISR-driven changes in gene expression. This transcriptional rewiring is vital for cells to prioritize and adapt their response to stress challenges (Anderson and Haynes, 2020; Forsstrom et al, 2019; Ost et al, 2020).

In mammals, mitochondrial stress triggers the ISR in concert with the UPR^mt. Noteworthy, the UPR^mt was first identified in mammalian cells, but then thoroughly characterized in yeast. There is still limited knowledge of the specific proteins and mechanisms governing the mammalian UPR^mt, but certain key pathways have been identified (Fig. 3). The canonical UPR^mt in mammals mirrors what happens in *C. elegans* and involves Activating Transcription Factor 5 (ATF5), the mammalian homolog of the protein ATFS-1 in *C. elegans*. In unstressed cells ATF5 is transported from the cytosol and into the mitochondria where it is degraded by Lon Peptidase 1 (LONP1), however under mitochondrial stress conditions ATF5 is blocked from entering the mitochondria and is

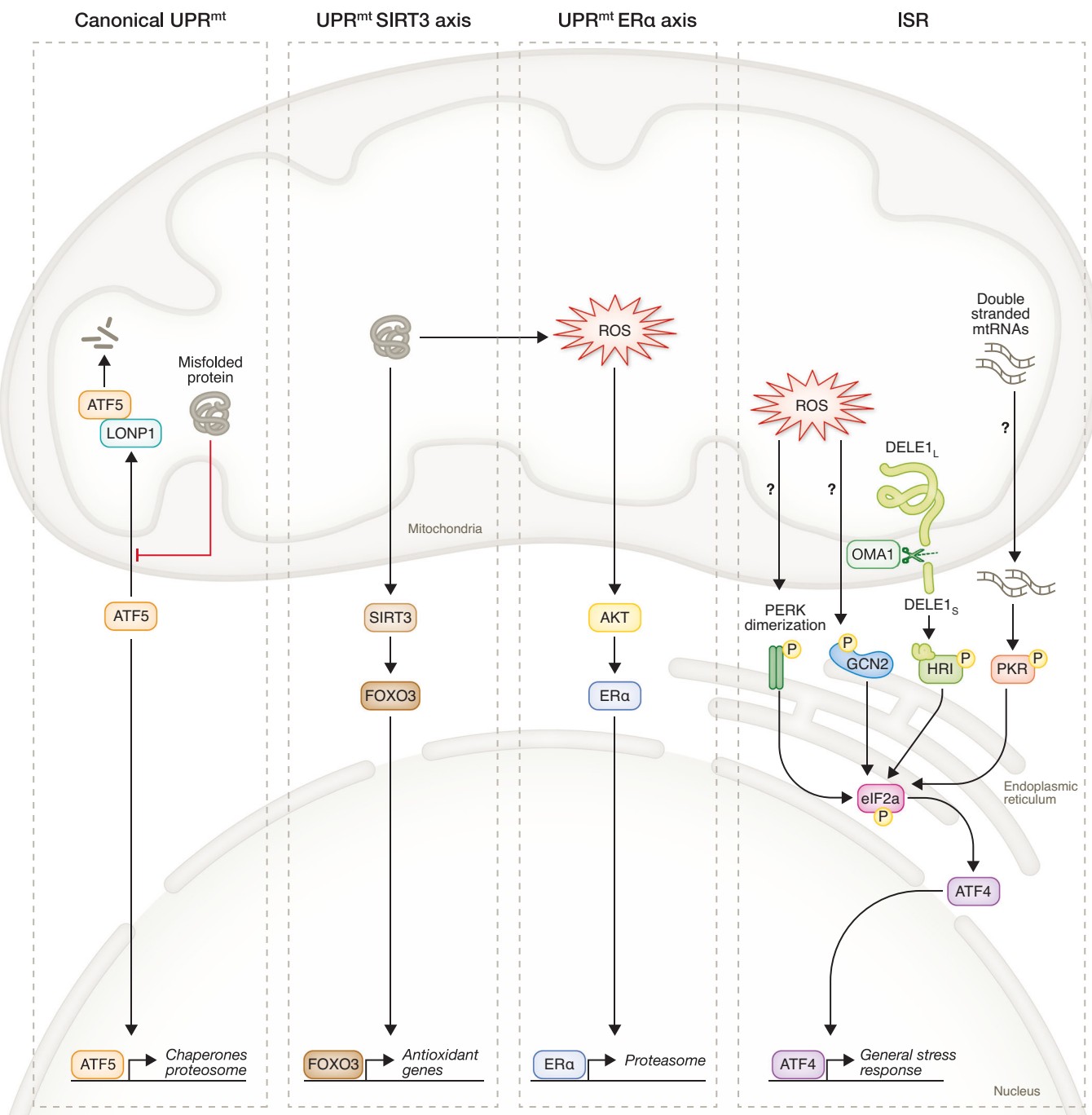

**Figure 3.  Key mitochondrial stress response pathways in mammalian cells.**

In mammals the mitochondrial unfolded protein response (UPRmt) and integrated stress response (ISR) are concomitantly upregulated in response to mitochondrial stress. The canonical UPRmt is conserved between mammals and *C. elegans* and involves the protein ATF5. Other noncanonical arms of the UPRmt also exist in mammals, including the SIRT3 and ERα axes. Induction of the ISR during mitochondrial stress can occur via various pathways. The stress kinases PERK, GCN2, HRI, and PKR have all been implicated in mitochondrial integrated stress response (ISRmt) activation and converge on the crucial phosphorylation of eIF2α and downstream increase in ATF4 levels. However, there are still unknowns in the specific mechanisms of how each stress kinase is activated.

translocated into the nucleus where it increases chaperone and proteasome gene expression to restore mitochondrial homeostasis (Anderson and Haynes, 2020; Fiorese et al, 2016; Pakos-Zebrucka et al, 2016). Other noncanonical axes of the UPRmt have also been characterized in mammals and can be upregulated concomitantly.

SIRT3 has been shown to increase in expression under mitochondrial stress and acts to deacetylate the transcription factor Forkhead box O3 (FOXO3) to increase transcription of antioxidant genes (Papa and Germain, 2014; Tseng et al, 2013). Estrogen Receptor Alpha (ERα) has also been implicated to play a role in the

mammalian mitochondrial stress response, although this pathway is less characterized. A study found that ERα increases proteasome expression in the intermembrane space (IMS) upon mitochondrial stress (Papa and Germain, 2011), while another showed that in female mice ERα is indispensable for the maintenance of mitochondrial function in muscle (Ribas et al, 2016).

ISR^mt has also been shown to progress in stages in the mammalian system, with Fibroblast growth factor 21 (FGF21) playing a key role in the progression of the ISR^mt (Forsstrom et al, 2019). In the study, the authors use the term ISR^mt to refer to how, in mammals, the response to mitochondrial stress does not just involve the mammalian homologs of proteins involved in the UPR^mt that have been characterized in invertebrates, but also includes stress response factors involved in the canonical ISR pathway (Khan et al, 2017; Nikkanen et al, 2016). This was described in the muscle of both mice and humans experiencing mitochondrial DNA (mtDNA) stress. The first stage of ISR^mt is characterized by upregulation in gene expression of FGF21, growth/differentiation factor 15 (GDF15), and ATF5. In stage 2, FGF21 acts locally to increase serine and glutathione (GSH) biosynthesis and acts systemically via cross-talk with various peripheral tissue types to affect whole-body metabolism. The third and final stage involves upregulation of more canonically understood UPR^mt genes such as mitochondrial heat shock proteins (HSPs) and *Atf3*.

At the heart of the signaling cascade that triggers mitohormesis are various mitochondrial effectors or outputs, which differ based on the tissue type and the specific stress evoked. Among these, mitokines, signaling molecules released by mitochondria, play a crucial role in mediating systemic adaptations to mitochondrial stress. Notably, FGF21 and GDF15 have been well-studied for their ability to influence energy regulation and glucose homeostasis (Jena et al, 2023). Yet, there are other less characterized outputs of the ISR, including mitochondrial peptides, metabolites, and small extracellular vesicles (sEVs), which can transport diverse cargo such as miRNAs, mRNA, DNA, proteins, and metabolites. sEVs have also been shown to carry mitochondrial components. These specific sEVs are termed mitochondrial-derived vesicles (MDVs) and enable intercellular mitochondrial transfer between tissues. (Crewe et al, 2021; Neuspiel et al, 2008) (Fig. 2A).

Overall, mitochondrial stress in mammals elicits a coordinated response via the upregulation of the ISR and UPR^mt and downstream secretion of signaling molecules that can influence systemic metabolism. Pioneering studies in the field of mitochondrial stress and mitohormetic mechanisms were performed in invertebrates; however it is crucial to elucidate the differing mechanisms that occur in the mammalian system to understand which mechanisms are crucial for an adaptive/mitohormetic response.

# The complexity of mammalian mitohormesis: from organelles to tissues cross-talk

The potential to treat and prevent a variety of diseases, ranging from metabolic disorders to the impacts of aging, by harnessing mitohormetic mechanisms is promising. Uncovering these biological pathways could lead to significant breakthroughs that could be

leveraged to treat diseases. Thus, there is a growing interest to expand research in this area and unlock its full potential, but first, we must gain a better understanding of the complex pathways involved within the current, established mammalian models that trigger beneficial rather than detrimental responses to stress stimuli. Mitochondrial stress can cause beneficial outcomes in the specific tissue type where the stress is evoked, systemic whole-body outcomes, or benefits to distal tissues, as a result of interorgan communication (Fig. 4). Here, we will discuss the latest mammalian mitohormetic models from the last 10 years that elucidate these implications of mitohormesis. These models are summarized and listed in Table 1.

## Tissue-specific, local mitohormetic models

In the mammalian models illustrated below, the authors trigger tissue-specific mitochondrial stress, targeting essential mitochondrial proteins involved in either the oxidative phosphorylation (OXPHOS) machinery or mitochondrial structure dynamics. The local tissue type attempts to resolve this mitochondrial stressor via activation of cellular stress pathways, leading to a transcriptional rewiring of cells. Although some detrimental results of the mitochondrial stress may still remain, upregulation of stress response factors can exert a beneficial effect on the tissue type experiencing the initial stressor and sometimes confer resilience to subsequent mitochondrial stressors. In this discussion, we explore the common pathways that tend to be upregulated in response to localized mitochondrial stress, as well as the downstream effects that these protective mechanisms produce. We acknowledge that many studies have primarily focused on identifying local mitohormetic stress without investigating potential systemic effects. Therefore, it's important to note that some models described here might have systemic effects in addition to the local ones observed.

An essential adaptive response that significantly contributes to alleviating mitochondrial stress is the boosting of the antioxidant response. This response is crucial for maintaining redox homeostasis during cellular stress reactions, ultimately leading to several beneficial outcomes. In the study by Ahola et al, the authors explore the effects of disrupting the electron transport chain (ETC) by knocking out complex IV assembly factor Cox10 (heme A:farnesyltransferase) in the heart and muscle tissues of mice (Ahola et al, 2022). Despite the severe mitochondrial dysfunction leading to ISR activation, the heart was protected from ferroptosis, a form of iron-dependent cell death driven by lipid peroxidation (Stockwell et al, 2017). In this animal model the ISR was activated through the Oma1–DELE1-ATF4 pathway (Fessler et al, 2020; Guo et al, 2020). This pathway is a key mechanism that allows mitochondrial stress to activate the ISR in mammalian cells. Activation of the ISR led to the increased expression of enzymes in the trans-sulfuration pathway, which facilitate GSH synthesis and the incorporation of selenium into selenoproteins like Glutathione Peroxidase 4 (GPX4) in mouse embryonic fibroblasts (MEFs). This resulted in increased levels of GPX4, a protein that utilizes GSH to reduce lipid peroxide radicals to prevent ferroptosis (Yang et al, 2014). Antioxidant-related genes were also upregulated, such as Nuclear factor erythroid 2-related factor 2 (*Nrf2*), and GSH metabolism enzymes. These effects resulted in protection from lipid peroxidation in the Cox10^-/- heart and resistance to ferroptosis.

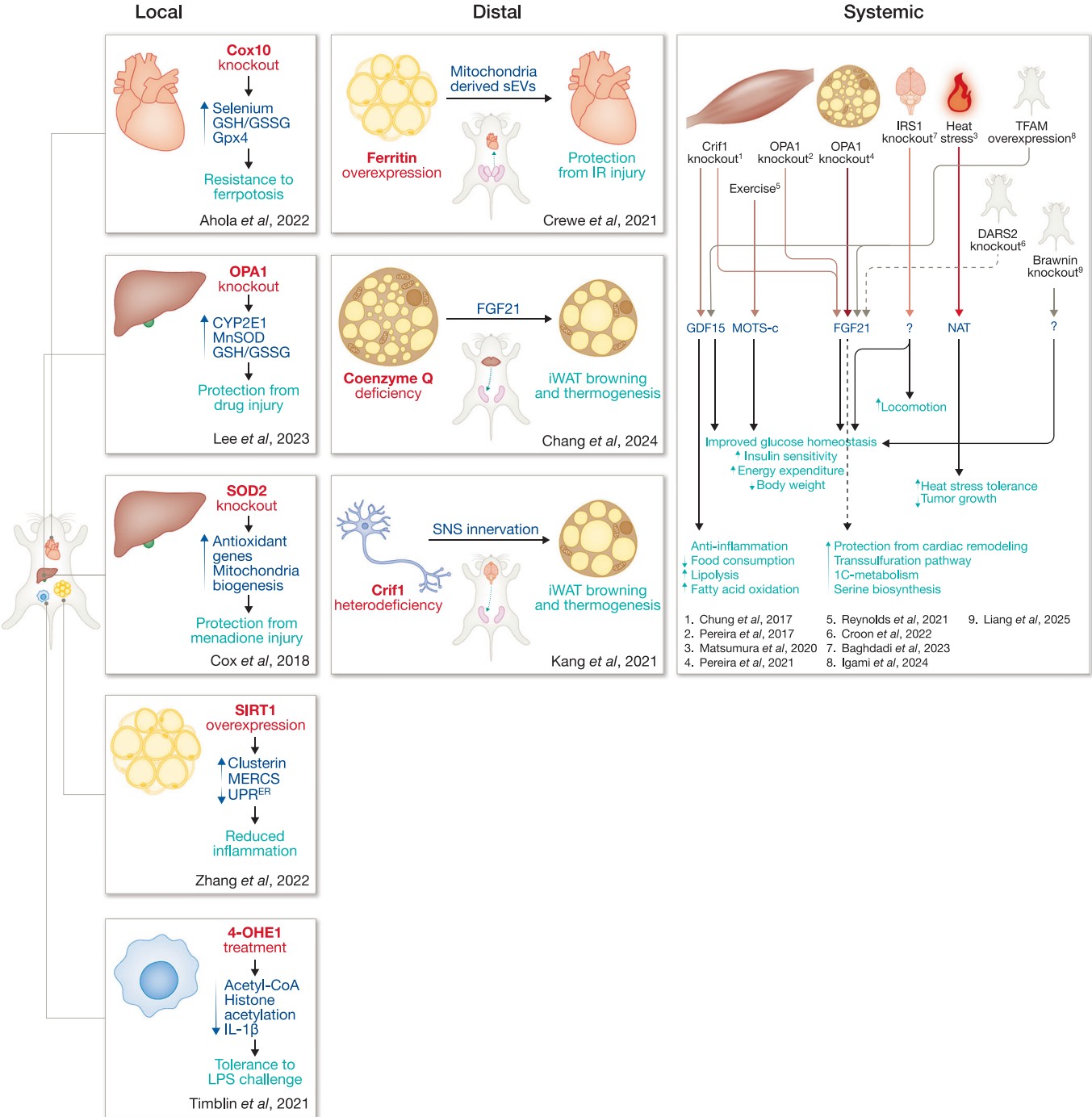

**Figure 4. Mammalian models of local, distal, and systemic mitohormesis.**

Recent studies have uncovered mechanisms of mitohormesis in mouse models. In these studies, mitochondrial stress signaling causes beneficial local, distal, and systemic outcomes.

Upregulation of the antioxidant response can lead to a variety of local outcomes, beyond just preventing cell death. Recent studies have shown that when mitochondrial stress prompts an upregulation of antioxidant proteins, it can shield different tissues prone to oxidative stress, like the liver. Given that the liver coordinates systemic metabolism, the functionality of its mitochondria is crucial for maintaining metabolic flexibility throughout the body. In a study by Lee et al, mitochondrial stress caused by a liver-specific Optic Atrophy 1 (OPA1) knockout (OPA1^LKO) initiated an antioxidant response, ultimately resulting in protection from drug-induced liver injury (Lee et al, 2023). OPA1 is a key regulator of mitochondrial fusion (Cipolat et al, 2004), cristae remodeling

**Table 1. Relevant mammalian mitohormesis models.**

| Reference | Trigger of mitochondrial stress | Tissue(s) experiencing stress | Local, systemic or distal effect | Identified mitohormetic effector | Mitohormetic outcome |
|---|---|---|---|---|---|
| Ahola et al, 2022 | Cox10 knockout | Skeletal muscle and heart | Local | N/A | Ferroptosis resistance |
| Lee et al, 2023 | OPA1 knockout | Liver | Local and systemic | N/A | Liver protected from drug-induced injury, improved glucose tolerance, increased oxygen consumption and decreased weight gain |
| Cox et al, 2018 | Transient SOD2 knockdown | Whole body, but focusing on liver phenotype | Local | N/A | Upregulation of antioxidant response and mitochondrial biogenesis |
| Zhang et al, 2022 | SIRT1 overexpression | Adipose tissue | Local and systemic | N/A | Reduced inflammation in obese adipose tissue leading to improved insulin and glucose tolerance, lower body weight, increased energy expenditure |
| Timblin et al, 2021 | 4-OHE1 treatment | Macrophages | Local | N/A | Tolerance to LPS challenge and subsequent secondary challenge |
| Pereira et al, 2017 | OPA1 knockout | Skeletal muscle | Systemic | FGF21 | Leaner phenotype, improved glucose and insulin |
| Pereira et al, 2021 | OPA1 knockout | Brown adipose tissue (BAT) | Systemic | FGF21 | iWAT Browning, improved cold tolerance, resistance to DIO and insulin resistance |
| Croon et al, 2022 | CLPP knockout | Whole body | Systemic | FGF21 | iWAT Browning, lower body weight, protection from cardiac remodeling and cardiomyopathy |
| Reynolds et al, 2021 | Exercise | Whole body | Systemic | MOTS-c | Improved physical capacity, enhanced energy metabolism, delayed aging-related phenotypes |
| Chung et al, 2017 | Crif1 knockout | Skeletal muscle | Systemic | GDF15 | Improved insulin and glucose tolerance, white fat mass reduction, increased lipolysis and fatty acid oxidation |
| Kang et al, 2021b | Crif1 knockout | liver | Systemic | FGF21 and GDF15 | FGF21 influences thermogenesis and energy expenditure while GDF15 improves hepatic steatosis, body weight and composition |
| Matsumura et al, 2020 | Heat or restraint stress | Whole body | Systemic | NAT | Tolerance to heat and restraint stress, inhibition of tumor growth |
| Baghdadi et al, 2023 | IRS1 knockout | Neurons | Systemic | N/A | Increased locomotion, insulin sensitivity, and energy expenditure |
| Liang et al, 2025 | Complex III knockout | Whole body | Local and systemic | N/A | Improved mitochondrial capacity within heart (i.e., reduced ROS and increased FAO). Systemic increase in energy expenditure and RER during exercise. |
| Crewe et al, 2021 | Iron deficiency by ferritin overexpression | Adipose tissue | Distal signaling to heart | Mitochondria-derived sEVs | Improved phenotype post IR injury in heart |
| Chang et al, 2024 | Coenzyme Q deficiency by PDSS2 knockout | Brown adipose tissue | Systemic outcomes stem from distal signaling to iWAT | FGF21 | Increased iWAT browning resulting in resistance to DIO, increased oxygen consumption, and decreased fat composition |
| Kang et al, 2021a | Crif1 heterozygous knockout | POMC neurons | Systemic outcomes stem from distal signaling to iWAT | MOTS-c and β-END | Propagation of UPR$^{mt}$, increased iWAT innervation, iWAT browning, increased energy expenditure |
| Igami et al, 2024 | TFAM overexpression | Whole body | Systemic | FGF21, GDF15 | Longer lifespan, leaner mice, smaller in size |

(Frezza et al, 2006), and supercomplex assembly (Cogliati et al, 2013). In OPA1$^{LKO}$ mice, loss of OPA1 resulted in proteostasis stress, triggering the ISR and UPR$^{mt}$ activation. However, when exposed to acetaminophen (APAP) overdose, OPA1$^{LKO}$ animals displayed no liver injury. Authors found that upregulation of the ISR and UPR$^{mt}$ caused a transcriptional rewiring resulting in decreased expression of Cytochrome P450 2E1 (CYP2E1), the protein involved in converting APAP to the toxic reactive metabolite N-acetyl-p-benzoquinone imine (NAPQI). This, combined with enhanced levels of antioxidant enzyme Manganese Superoxide Dismutase (MnSOD) and GSH/glutathione disulfide

(GSSG) ratio, allowed prevention of harmful oxidants from being formed, and promotion of scavenging of existing ROS. Similar to the proteostasis stress in the aforementioned study (Lee et al, 2023), priming organisms with mild oxidative stress can result in protection from subsequent challenges. Similar protective effects have been observed in a liver-specific Superoxide dismutase 2 (SOD2) knockdown model, where upregulation of antioxidant response genes safeguarded cells from oxidative stress caused by pro-oxidant drugs such as menadione (Cox et al, 2018). Interestingly, SOD2 is known to scavenge oxygen radicals, however, an inducible and reversible knockdown in mice (iSOD2-KD) using a

system with rtTA expressed from the Rosa26 promoter, caused a compensatory response. Using this model, they induced an embryonic 4-day knockdown of SOD2. The livers of iSOD2-KD mice exhibited upregulation of antioxidant genes such as heme oxygenase-1 (HO-1) (*Hmox1*), Glutamate-cysteine ligase regulatory subunit (*Gclm*), NAD(P)H quinone dehydrogenase 1 (*Nqo1*), and glutaredoxin-1 (*Glrx*) and mitochondrial biogenesis. This transcriptional rewiring was dependent on NRF2 and peroxisome proliferator–activated receptor gamma (PPARγ)/peroxisome proliferator-activated receptor gamma coactivator 1 alpha (PGC1α) activation. Upregulation of this compensatory response caused primary MEFs isolated from these mice, to be resistant to cell death when challenged with menadione. This shows that mild and transient oxidative stress can be protective against future oxidative stressors.

Notably, mitochondrial stress in the liver is a downstream outcome indicative of metabolic dysfunction associated with Metabolic dysfunction-Associated Steatotic Liver Disease (MASLD) in patients. Nutrient stress can cause mitochondrial overload and bioenergetic dysfunction, leading to increased levels of ROS and the production of inflammatory factors that cause the pathogenesis and progression of MASLD to Metabolic dysfunction-Associated Steatohepatitis (MASH) (Koliaki et al, 2015; Pérez-Carreras et al, 2003). In this disease progression, mitochondrial stress is chronic and exceeds the hormetic zone. The local tissue response to the stress is not enough to resolve the energetic dysfunction, and thus novel therapeutic approaches that target and resolve mitochondrial stress or lower ROS are being explored to treat MASLD. These therapeutics include liver-targeted mitochondrial uncouplers (Kanemoto et al, 2019; Perry et al, 2013; Perry et al, 2015) as well as PPAR agonists that can increase mitochondrial β-oxidation to relieve free fatty acids (FFAs) overload and thus mitochondrial stress (Cooreman et al, 2024; Francque et al, 2021; Wettstein et al, 2017). Despite these discoveries, mitohormesis inducers have not been thoroughly explored in relation to the treatment of MASLD.

Most relevant, it seems that the metabolic dysfunction associated with obesity can be improved via mitohormesis through upregulation of anti-inflammatory pathways. In the following study, mitochondrial stress triggers a compensatory mechanism that can mitigate the low-grade inflammation typically associated with obesity. In obese adipose tissue, the ER faces metabolic overload via the accumulation of FFAs, which can generate ROS and cause induction of the UPR^ER. UPR^ER activation leads to downstream upregulation of inflammatory pathways via c-Jun N-terminal kinase (JNK) and Nuclear Factor kappa-light-chain-enhancer of activated B cells (NF-κB) activity (Kawasaki et al, 2012). Mitohormetic mechanisms have been shown to help in resolving the chronic inflammation and lipotoxicity associated with obesity in the following study. A model of local mitochondrial stress was utilized by overexpressing Sirtuin 1 (SIRT1), a NAD+ dependent protein deacetylase, in an adipose tissue-specific manner (Adipo-SIRT1) (Zhang et al, 2022). Other studies have described that SIRT1 contributes to mitochondrial biogenesis in adipose tissue (Majeed et al, 2021). In the following study, overexpression of SIRT1 caused activation of the UPR^mt within Adipo-SIRT1 white adipose tissue (WAT). Downstream of UPR^mt upregulation, the expression of the mitochondrial chaperone Clusterin increased. Interestingly, upregulated Clusterin expression caused increased mitochondria-ER contact sites (MERCs) where Clusterin and

SIRT1 were colocalized. Although the mechanisms are still not fully elucidated, increasing MERCs in this model was able to decrease activation of UPR^ER, leading to less inflammation and improved whole-body metabolic parameters in a high-fat diet (HFD) model.

Just like adipocytes, macrophages play a vital role in regulating chronic inflammation responses in the body. As mentioned previously, pathogens represent a trigger that could elicit specific mitochondrial responses. Timblin et al tackle the question of whether a mitochondrial contribution to pathogen infection impacts macrophage functions (Timblin et al, 2021). The authors were interested in understanding how estrogens could affect immune function and found that 4-hydroxyesterone (4-OHE1) caused decreased adipose tissue inflammation within an obese mouse model and decreased Interleukin-1 beta (IL-1β) levels in serum in mice challenged with lipopolysaccharide (LPS) stimulation. They concluded that 4-OHE1 treatment caused covalent adducts with mitochondrial proteins, resulting in activation of stress response pathways and downstream disruption of metabolite pools, notably CoA. Because Acetyl-CoA is required for histone acetylation of pro-inflammatory genes (Lauterbach et al, 2019), lowered levels led to decreased inflammation-related gene expression. This rendered macrophages tolerant to both primary and secondary LPS challenges. This study reveals a mechanism whereby mitochondrial stress responses prime macrophages by causing changes to metabolites, propagating anti-inflammatory outcomes in response to LPS challenge. Targeting this pathway downstream of Toll-like receptor (TLR) activation can be used as a therapeutic strategy.

In summary, these papers emphasize the distinct mechanisms by which inducing mitochondrial stress can result in local beneficial outcomes within the tissue or cell type experiencing that stress. Targeting essential mitochondrial proteins, which are important for bioenergetic or structural regulation, can trigger UPR^mt and/or ISR^mt, causing activation of compensatory pathways and protection from subsequent stressors. In addition, downstream upregulation of antioxidant genes, including NRF2-target genes and those involved in GSH metabolism, induced by UPR^mt, represents a conserved way tissues can recover from stress or become resistant to subsequent oxidative insult. These studies demonstrate that anti-inflammatory responses initiated by the UPR^mt can provide both localized and systemic advantages, particularly in tissue types that play a crucial role in triggering an organism's inflammatory reaction to various stressors. It appears that these anti-inflammatory responses are quite intricate and may not be solely dependent on a specific set of key genes. Rather, they seem to fluctuate according to the metabolic demands of the tissue under stress.

## Systemic mitokine-induced effects of mammalian mitohormesis

In numerous studies centered on mitohormesis, a mitochondrial stressor leads to the activation of the ISR or UPR^mt, ultimately resulting in the secretion of a factor that can confer systemic, whole-body benefits. The following section highlights whole-body metabolic outcomes that occur downstream of mitochondrial stress. In the following section, we will describe recent studies that contain key findings on well-known as well as newly discovered

mitokines in mitohormetic models. We will also discuss the specific systemic metabolic benefits that these downstream signaling molecules exert and the overlapping or differing metabolic benefits that occur as a result of each mitokine.

## FGF21

FGF21 is a well-characterized mitokine that has been identified to play a role in mitohormesis as a novel metabolic regulator (Kharitonenkov et al, 2005). Increased FGF21 levels cause improved glucose homeostasis and weight loss (Xu et al, 2009) and thus it has been explored as a potential therapeutic for metabolic diseases such as MASLD and Type 2 Diabetes Mellitus (T2DM) (Bhatt et al, 2023; Cui et al, 2020; Loomba et al, 2023). Recent studies have revealed that the upregulation of FGF21 secretion in response to mitochondrial stress plays a crucial role in how organisms adapt and compensate to mitochondrial stress (Tyynismaa et al, 2010). In two studies conducted by Pereira et al, FGF21 is identified as the mitokine responsible for triggering systemic metabolic changes. Rather than pinpointing a single target tissue, the authors propose that various peripheral organs orchestrate the overall mitohormesis effects observed. In the mammalian models proposed, the authors induce mitochondrial stress by using tissue-specific knockouts of the mitochondrial protein OPA1 in tissues with high bioenergetic demands, such as skeletal muscle (Pereira et al, 2017) and BAT (Pereira et al, 2021). In both cases, the abrogation of OPA1 leads to altered mitochondrial morphology and triggers the ISR, resulting in the secretion of FGF21 from muscle and brown adipose tissue (BAT), respectively. Muscle-derived FGF21 causes improved insulin sensitivity and glucose regulation in aged and obese mice. However, to understand the dependence on FGF21 in the BAT-specific KO model, the authors use an OPA1 FGF21 double knockout (DKO) mouse and conclude that the improved glucose metabolism and resistance to diet-induced obesity (DIO) phenotype happens through FGF21-independent mechanisms. This suggests there may be another mitokine responsible for these metabolic changes in OPA1 BAT KO mice. They also recognized a key mechanism whereby ATF4 upregulation due to ISR induction is necessary for FGF21 upregulation and secretion from BAT. A recent study by Croon et al demonstrates that FGF21 acts as a mitokine with stress-intensity–dependent effects in the heart and a key modulator of ISR$^{mt}$ induction (Croon et al, 2022). Using a model of severe mitochondrial dysfunction, mitochondrial aspartyl-tRNA synthetase (DARS2) KO mice, the authors show that profound OXPHOS deficiency failed to engage FGF21-mediated protective signaling. In contrast, in a milder mitochondrial stress model (mitochondrial matrix CLPP protease knockout), FGF21 promotes adaptive responses by activating ERK1/2 signaling and directly regulating components of the mitochondrial integrated stress response (ISR$^{mt}$), thereby protecting the heart from early pathological remodeling and cardiomyopathy. Notably, FGF21 control of the ISR$^{mt}$ and one-carbon metabolism appears cardiomyocyte-specific, as FGF21 depletion in skeletal muscle or adipose tissues (BAT, WAT) did not affect expression of one-carbon metabolism enzymes. These findings highlight a pattern commonly seen in ISR signaling, clearly demonstrating how the context- and tissue-specific nature of mitokine-driven mitohormesis allows systemic mitokine signaling to produce finely tuned, organ-specific adaptive responses to mitochondrial stress.

## GDF15

GDF15 is another well-characterized mitokine that is upregulated downstream of the UPR$^{mt}$ and ISR activation. Like FGF21, it influences systemic metabolism and often contributes to a healthier and leaner phenotype in vivo. While the exact mechanism of action is still not completely understood, research has indicated that it possesses anti-inflammatory properties, reduces food intake, and boosts metabolism (Abulizi et al, 2017; Emmerson et al, 2017; Johnen et al, 2007). Chung et al and Kang et al explored this area of interest by utilizing a mammalian model with depletion of mitochondrial ribosomes (mitoribosomes). These specialized ribosomes play a crucial role in translating the components of OXPHOS that are encoded in the mitochondrial genome. The authors developed both muscle (Chung et al, 2017) and liver-specific knockouts of Crif1 (CR6-interacting factor 1) gene (Kang et al, 2021b). Interestingly, both models displayed improvement of insulin sensitivity and reductions in fat mass or body weight. RNA-seq analysis also revealed high circulating levels of both GDF15 (Growth Differentiation Factor 15) and FGF21. While both mitokines were found to be elevated in Crif1-deficient mice, the study specifically focused on GDF15 and its essential role in metabolic adaptation. Administering GDF15 to mice led to increased lipolysis and fatty acid oxidation, mirroring the metabolic benefits observed in mouse models expressing human GDF15. In the liver-specific study, the authors outlined the different functions of GDF15 and FGF21 in response to mitochondrial stress. Although both secreted factors positively impact whole-body metabolism, a double knockout of Crif1 and Gdf15 genes demonstrated that GDF15 helps improve hepatic steatosis, body weight, and fat composition. In contrast, a double knockout of Crif1 and Fgf21 revealed that FGF21 specifically affects thermogenesis and energy expenditure. This study strongly suggested how these mitokines played a separate role in enhancing insulin sensitivity. These findings also highlight the therapeutic potential of FGF21, GDF15, or their analogs in managing obesity-related diseases, particularly T2DM, with possible clinical application, as shown in the first clinical trial in obese humans (Benichou et al, 2023; Breit et al, 2023).

## MOTS-C

One of the less characterized signaling factors downstream of ISR activation is the mitochondrial open reading frame of the 12S rRNA type C (MOTS-c). Short open reading frames (sORFs) in the mitochondrial genome have been recognized to produce bioactive peptides called mitochondrial-derived peptides (MDPs) with diverse physiological functions. Recent papers have identified MOTS-c as a novel MDP that regulates muscle metabolism, insulin sensitivity, and weight regulation (Lee et al, 2015). MOTS-c regulates glucose homeostasis by interplaying the folate cycle, 5-aminoimidazole-4-carboxamide ribonucleotide (AICAR), and AMP-activated protein kinase (AMPK) signaling pathways. Moreover, it shares similar physiological effects with therapeutic agents such as methotrexate and metformin, indicating its potential as a therapeutic target for metabolic disorders such as T2DM.

The positive outcomes associated with exercise have been recognized to be a result of exercise's ability to induce mitohormesis, and MOTS-c has been implicated in this pathway (Ristow et al, 2009). This was highlighted in a recent paper (Reynolds et al, 2021) where the authors first assessed whether MOTS-c levels respond to

physical exercise. In human volunteers, post-exercise MOTS-c levels were significantly increased, suggesting its role in exercise adaptation. In mice, MOTS-c treatment enhanced physical capacity and running performance, improving whole-body energy metabolism independent of body weight. The study also proposed that MOTS-c may promote longevity by maintaining cellular homeostasis, delaying age-related physical decline, and reducing morbidity in later life. Furthermore, RNA-seq analysis revealed that MOTS-c regulates pathways related to proteostasis, including protein regulation and metabolism, indicating its upregulation in response to metabolic and oxidative stress. Despite these promising findings, additional research is needed to elucidate the downstream pathways regulated by MOTS-c before it can be considered for therapeutic applications to enhance healthspan and longevity.

### NAT

Another interesting mitohormetic factor that was recently identified by Matsumura et al, is N-acetyle-l-tyrosine (NAT) (Matsumura et al, 2020). They first identified NAT through studies in a non-mammalian organism. They described a specific peptide, NAT, present in parasitized larvae, which was able to induce heat stress tolerance in armyworm larvae in a time-dependent manner after injection. Even though there had been no previous reports of the presence of NAT in mammals, the authors detected NAT in human serum and observed increase NAT concentrations in mice after heat and restraint stress, suggesting it may represent an endogenous metabolite and hormesis inducer in mammals as well. The authors found a mechanism, in *Drosophila*, whereby NAT transiently perturbed mitochondria, promoting mitochondrial reactive oxygen species (mROS) production. NAT also increased Kelch-like ECH-associated protein 1 (*Keap1*) transcription levels via the FoxO–Keap1 signaling axis activation. Elevated KEAP1 protein inhibited NRF2, thereby suppressing NRF2 -target antioxidant enzyme genes and enhancing mROS production and activating mitochondrial stress responses. However, in the second stage of this mechanism, NRF2 can evade Keap1-mediated repression and enhance expression of antioxidant enzyme genes, leading to improved stress tolerance. Interestingly, since NRF2 is aberrantly activated in some cancers, NAT supplementation significantly inhibited the growth of colorecal cancer cells in nude mice, mostly due to the activation of FoxO. Although, the specific mechanism by which NAT suppresses tumor growth through mROS generation remains to be elucidated, this study identified a novel mitohormetic factor. Further studies are warranted to uncover other unknown mediators of mitohormesis.

### IRS1 Deletion and Neuronal Stress Signaling

One of the biggest challenges in exploring these systemic effects is the difficulty of establishing a clear cause-and-effect relationship between individual secreted factors and alterations in overall body metabolism. A recent study was able to identify that whole-body deletion of insulin receptor substrate 1 (IRS1) resulted in increased lifespan and healthspan through mitochondrial stress activation in neurons (Baghdadi et al, 2023). They found a reshaping of mitochondrial function and indicators of mitochondrial stress, such as activation of ATF4. This led to systemic outcomes such as increased locomotion, insulin sensitivity, and energy expenditure in old male mice, but not female mice. Intriguingly, they also found that mammalian mitochondrial ISR signaling propagated from the

brain to peripheral organs. While they were able to conclude that this phenotype occurs independently of FGF21, they failed to identify the signaling factor responsible for this phenotype. This study highlights one of the most pressing gaps in our knowledge of mitohormesis. Beyond the conventional suspects (i.e., FGF21, GDF15), there exists a plethora of unknown secreted factors that impact whole-body metabolism in response to mitochondrial stress.

### Brawnin and Respiratory Supercomplexes

It is frequently difficult for researchers to characterize the complex mechanisms downstream of mitochondrial stress because the UPR$^{mt}$ still lacks characterization in the mammalian system. For instance, a recent study by Liang et al, presents a mammalian model of mitohormesis driven by alterations in respiratory supercomplexes (Liang et al, 2025). The gene encoding Brawnin, a small inner mitochondrial membrane protein essential for the assembly of ETC complex III (CIII), was deleted to generate a knockout (BR KO) mouse model. Surprisingly, despite defects in CIII assembly, mitochondria-dense tissues such as the heart, displayed improved mitochondrial function. This was attributed to the formation of larger ETC supercomplexes, named SC-XL. These adaptations included reduced ROS production, sustained respiratory function, and increased fatty acid oxidation. On a systemic level, indirect calorimetry revealed that BR KO mice exhibited higher energy expenditure than their wild-type (WT) counterparts and a higher respiratory exchange ratio (RER) during exercise. While the authors describe unexpected benefits downstream of CIII activity reduction, the specific signaling cascade that starts at the mitochondrial defect level and results in systemic benefits is difficult to parse out without more characterization of the mammalian mitochondrial stress response. Interestingly, the study also underlined a key divergence from previous findings in zebrafish, where BR KO resulted in lethal mitochondrial disease (Zhang et al, 2020). This divergence highlights how similar mitochondrial disturbances can produce varying outcomes across species, emphasizing the intricate nature of mitochondrial resilience and adaptation in different models.

All these observations, taken together, show the multifaceted aspects of ISR and its activation. Mitokines are often secreted in concert during mitochondrial stress, and different stressors could have similar patterns and outcomes to elicit systemic effects. Identifying the mechanisms as well as specific mediators or tissues contributing to these effects can be difficult, leading to significant challenges in understanding the impacts of specific stressors, especially when multiple organs contribute to orchestrate the overall systemic effects observed. However, the continuous development of new technologies and tools could help us better understand and define the complex mechanisms and interplay underlying systemic mammalian ISR dynamics.

## Interorgan communication in mitohormesis models: when tissues send SOS signals

When the local stress response cannot resolve mitochondrial stress in a specific tissue, signaling factors secreted from these tissues recruit distal tissues in order to cope with such stress. This phenomenon was first discovered in *C. elegans* (Durieux et al, 2011); however, a small handful of papers recently indicated that similar cell non-autonomous communication occurs in mammals,

which ultimately causes mitohormesis. The studies summarized in this section differ from those mentioned above because they describe specific interorgan communication between a specific tissue type experiencing stress and a peripheral compensating tissue. Pathways that are upregulated in compensatory tissues ultimately lead to beneficial metabolic outcomes or protection from injury.

Crewe et al described the first mammalian interorgan mitohormesis model by utilizing a model whereby ferritin, an iron chelator, was overexpressed in an adipose tissue-specific manner (adipo-FtMT), causing iron deficiency within adipose tissue (Crewe et al, 2021). Due to the importance of iron and heme in the mitochondrial ETC, adipose tissue of adipo-FtMT mice had increased levels of oxidative stress. The authors discover a unique signal being secreted from stressed adipose tissue in the form of mitochondria-derived small extracellular vesicles (sEVs). They also found that the secretion of sEVs by adipocytes and the uptake of these sEVs into the heart caused the propagation of oxidative stress. Further, the authors illustrate a mitohormetic mechanism whereby injection of sEVs from palmitate-supplemented adipocytes into mice prior to ischemia/reperfusion (I/R) injury was protective and resulted in less local tissue death, lipid peroxidation and improved heart function post-I/R. Thus, they uncovered an adipocyte to cardiomyocyte ROS transfer that caused cardiac resilience post-I/R. While this study provides one of the first models of mammalian interorgan mitohormesis involving mitochondrial transfer, it also emphasizes the broader role of interorgan communication as a protective response to mitochondrial stress.

The mechanisms of intercellular mitochondria transfer, as well as the implications this has for disease states, is a growing field (Borcherding and Brestoff, 2023). Mitochondrial transfer as a protective mechanism in mammalian cells was first described by Spees et al (Spees et al, 2006), and since then, groups have found that mitochondrial transfer happens among a variety of cell types and can cause protection from injury (Hayakawa et al, 2016; Islam et al, 2012). These studies highlight mitochondrial transfer via sEVs as an important signaling molecule for interorgan communication during mitohormesis.

Adipose tissue is increasingly recognized and studied as a crucial organ in the mitohormesis response in mammals. Besides its well-known function as a storage organ, research has highlighted its emerging role as a secretory organ, where it actively contributes to various physiological processes (Cypess, 2022; Deshmukh et al, 2019). Among these secreted proteins are adipokines, which are secreted upon mitochondrial stress. In a study by Chang et al, the secretory capacity of BAT during mitochondrial stress triggered by Coenzyme Q (CoQ) deficiency was characterized. Local BAT CoQ deficiency surprisingly led to a mitohormetic response involving interorgan communication (Chang et al, 2024). CoQ has long been appreciated as an essential cofactor for the transfer of electrons in the mitochondrial ETC and also as an antioxidant that protects cells from oxidative damage (Wang et al, 2024). Initial studies using a UCP1-cre driven knockout of the CoQ biosynthetic enzyme PDSS2 (PDSS2$^{BKO}$), identified that CoQ deficiency in BAT caused mitochondrial dysfunction, suppression of UCP1 expression, and cold intolerance in vivo (Chang et al, 2022). A more recent study, demonstrated that this downregulation of UCP1 expression is dependent on upregulation of the ISR and UPR$^{mt}$ response in BAT. This stress response activation is caused by a unique pathway in

which stressed mitochondria release mtRNAs into the cytosol, causing activation of the stress kinase PKR, as previously described (Sud et al, 2016). Unexpectedly, despite mitochondrial stress and BAT dysfunction, secretion of FGF21 from BAT resulted in enhanced whole-body respiration and resistance to diet-induced obesity. iWAT was identified as a key FGF21 target likely contributing to the mitohormetic phenotype. Interestingly, iWAT tissue pieces isolated from PDSS2$^{BKO}$ mice had increased browning as well as increased respiration rates. Thus, in the face of BAT thermogenic dysfunction that occurs downstream of mitochondrial stress, the organism tries to compensate by upregulating thermogenic pathways in iWAT, resulting in beneficial whole-body outcomes (Chang et al, 2024). This compensatory mechanism highlights how different adipose depots can interact non-autonomously to maintain systemic energy homeostasis. The compensatory mechanism highlighted in this study somewhat mirrors the work done by Pereira et al (Pereira et al, 2021). Both studies reveal that the ATF4-FGF21 axis could play a pivotal role in the stress adaptation coming from BAT; nevertheless Chang et al, identified the specific cross-talk between BAT and iWAT leading to the mitohormetic phenotype.

This research did not address whether FGF21 secreted by brown adipose tissue (BAT) acted directly on iWAT or indirectly through the central nervous system (CNS) to promote the mitohormetic phenotype. Certain studies have reported FGF21's direct action on adipose tissue (Samms et al, 2016), while others described that FGF21 acts on the CNS (Bookout et al, 2013). Kang et al describe a model whereby iWAT-induced browning was initiated through mitohormetic signaling factors secreted from the CNS (Kang et al, 2021a). This study emphasizes the role of the CNS in relaying various signals between communicating tissues and underscores iWAT as a key tissue in the compensatory response to mitochondrial stress. The authors utilize a proopiomelanocortin (POMC) neuron-specific deletion of Crif1. POMC neurons exist in the hypothalamus and are known to affect energy metabolism (Yaswen et al, 1999). While homozygous knockout mice had a maladaptive phenotype, the heterodeficiency model (Pomc-cre; Crif1 f/+) had protection from DIO and increased energy expenditure. Of note, both BAT and, to a larger extent, iWAT revealed metabolic changes, including increased thermogenic gene expression along with a propagation of UPR$^{mt}$ activation from POMC neurons. Increased innervation through the sympathetic nervous system (SNS) was shown to be driving the iWAT browning phenotype in Pomc-cre; Crif1 f/+ mice. Ultimately, the authors identify MOTS-c and β-END as downstream mediators of the POMC neuron to iWAT signal propagation. Previously, we discussed how MOTS-c is released during exercise (Reynolds et al, 2021). Interestingly, the authors were able to recapitulate the Pomc-cre; Crif1 f/+ phenotype through moderate-intensity training in mice, triggering ROS production in POMC neurons and increased hypothalamic MOTS-c of β-END expression. This showed that the POMC neuron to iWAT interorgan communication may be governing the mitohormetic benefits of exercise. Overall, this paper gave insight into how the CNS and SNS may be mediating the actions of signaling factors during mitohormesis.

These papers describe models in which tissues experiencing mitochondrial stress, recruit the help of peripheral organs to compensate. This results in organismal resilience to future insults or metabolic benefits. More research is necessary to uncover how

secretory factors contribute, either directly or indirectly, to changes in peripheral tissues. Remarkably, these studies highlight subcutaneous WAT as a key compensatory organ that receives disparate signals from stressed tissue to induce browning. Targeting subcutaneous WAT in models of mitochondrial stress may serve as a unique mechanism for resolving stress and also as a novel target for treatments of obesity, which is known to be a pathophysiological state with chronic levels of oxidative stress (Cheng et al, 2021; Furukawa et al, 2004).

## Mitohormesis as a target for treatment

As more studies have emerged highlighting mitochondria as an important signaling hub for resolving cellular stress, other studies are approaching the use of mitohormesis as a possible treatment by employing drugs that can induce mitohormetic effects and target key pathways involved in different diseases. In this section, we provide a broad overview of recent findings that illustrate how activating mitohormetic pathways in mammalian systems could serve as a promising approach for treating a variety of conditions, including but not limited to obesity, cancer, neurodegenerative diseases, and age-related decline.

The following studies reveal a paradoxical fact: both inhibiting or activating the ISR$^{mt}$ could have beneficial effects in disease models. This can be explained in the context of mitohormesis. In disease states with high levels of mitochondrial dysfunction and chronic ISR$^{mt}$ activation, treatment with an ISR inhibitor can reduce stress to within the hormetic range, resulting in beneficial effects. On the other end of the spectrum, when ISR$^{mt}$ activation is too low to allow compensation for a stressor, treatment with an ISR$^{mt}$ activator can raise stress signaling into the hormetic range (Fig. 2B). This balance highlights that there is no "one-size-fits-all" approach to treating mitochondria-related diseases. The disease type, stage of disease progression, and patient-specific responses must all be considered when modulating mitohormesis for therapeutic purposes.

### Obesity and related disorders

The reconfiguration of metabolic pathways and dysfunction in mitochondrial bioenergetics play significant roles in obesity and its associated conditions. Many of the pathological conditions associated with obesity are closely linked to mitochondrial dysfunction (de Mello et al, 2018). Key factors implicated in the progression of obesity-related disorders include impaired OXPHOS, increased reactive oxygen species (ROS) production, lipotoxicity, and reduced mitochondrial biogenesis, just to cite a few examples (Tung et al, 2024). These mitochondrial dysfunctions are generally considered consequences of common etiologies in obesity-related disorders, such as excessive nutrient intake, chronic inflammation, and lipid accumulation. Stressors like these activate stress responses, such as the ISR$^{mt}$ and the UPR$^{mt}$, which could potentially be leveraged to enhance mitochondrial resilience, improve metabolic function, and counteract the effects of obesity-related diseases, possibly even delaying the onset of obesity.

For example, metformin, widely used to treat T2DM, has pleiotropic effects, including mitohormetic ones. It has been linked to extended lifespan by increasing ROS production (De Haes et al,

2014), improved mitochondrial dysfunction, and regulated mitochondrial ETC complex levels and mitophagy. These mitohormesis pathways may partially explain the beneficial effects of metformin in obese and diabetic patients. A clinical trial (Coll et al, 2020) further explored metformin activation of mitohormesis and its implication in obesity. In a short-term human study, the authors identified increased circulating GDF15 levels in metformin-treated subjects, which significantly correlated with weight loss. To test this causality, they translated this study in mice, identifying that animals lacking GDF15 or its receptor GDNF family receptor alpha-like (GFRAL), did not experience metformin-induced weight loss. Metformin treatment has also shown cardioprotective effects in the context of myocardial I/R injury. In this context, metformin treatment preserved mitochondrial homeostasis and improved cardiac outcome (Tian et al, 2023). Severe myocardial injury typically leads to metabolic failure, leading to reduced ATP synthesis and causing excess ROS accumulation. This imbalance causes mitochondrial membrane potential depolarization, ultimately resulting in cell death (Brown et al, 2017). By increasing the AMP/ATP ratio and upregulating antioxidant enzymes such as SOD, metformin treatment, in this context, has protective effects against myocardial IR injury, improving mitochondrial function and reducing apoptosis.

Beyond metformin, other drugs that trigger mitohormesis by moderately impairing OXPHOS activity have been studied as treatments for obesity. Jiang et al provided a clear example of how targeting mitohormesis may yield beneficial metabolic effects (Jiang et al, 2024). The authors found that inhibitor of mitochondrial transcription (IMT) treatment induced paradoxical metabolic rewiring in HFD-fed mice, with positive effects on diabetic phenotype, such as decreased weight (fat mass) without affecting food intake, physical activity, or intestinal nutrient uptake. IMT also promoted a shift towards fatty acid oxidation at the organismal level, evidenced by decreased RER and impaired OXPHOS activities in the liver. The authors proposed that future studies using isotope-labeled substrates in isolated liver cells or whole animals will be necessary to strengthen this observation.

It remains unclear whether mitochondrial complications are secondary to the underlying drivers of metabolic diseases or what specific role mitochondrial quality control plays in this scenario. A recent study explored whether impairments in bioenergetics or quality control within mitochondria may act as contributing factors to the etiology of metabolic diseases, rather than being consequences of metabolic disorders such as T2DM (Walker et al, 2025). The authors proposed that dysfunctions in mitochondrial quality control activate a mitochondrial retrograde signaling program, such as the ISR, which contributes to the developmental immaturity of multiple metabolic tissues. The authors generated various mouse genetic models with deficiencies in components of the mitochondrial quality control machinery, such as the regulator of mitophagy and mitochondrial genome integrity for inducible β-cells deletion as well as liver-specific deletion. The study revealed that defects in mitochondrial quality control trigger a retrograde signaling pathway (mitonuclear signaling) across key metabolic tissues, including β-cells, hepatocytes, and brown adipocytes. Disruption in this signaling leads to a loss of cellular identity and maturity, resulting in compromised OXPHOS, activation of the ISR$^{mt}$, and chromatin remodeling. These mechanisms shift cells toward cellular immaturity rather than apoptosis,

ultimately contributing to metabolic dysfunction. Notably, the study also showed that pharmacological inhibition of ISR^mt in vivo, using the small molecule named integrated stress response inhibitor (ISRIB), originally identified by Sidrauski and colleagues in a cell-based screen targeting PERK activity (Sidrauski et al, 2013), could help restore β-cell identity and function, suggesting a promising therapeutic strategy for metabolic diseases such as T2DM.

By considering both mitohormetic interventions like metformin and approaches targeting mitochondrial retrograde signaling, future research could develop innovative treatments aimed at restoring mitochondrial quality control, preserving cellular identity, and improving metabolic homeostasis in obesity and related disorders.

## Cancer

Metabolic reprogramming and reshaping of mitochondrial pathways are common strategies used by cancer cells to sustain proliferation, invasiveness, and adaptation to the tumor microenvironment. As early as 1927, Otto Warburg discovered a link between mitochondrial dysfunction and cancer, termed the Warburg effect, noting that cancer cells tend to favor aerobic glycolysis over mitochondrial respiration for energy production (Warburg et al, 1927). Over the years, various other mechanisms of mitochondrial dysfunction have been identified in cancer cells, further deepening our understanding of this complex and multi-faceted disease. Different cancers tend to hijack and rewire ISR^mt and UPR^mt signaling pathways, either hyperactivating or suppressing stress responses to enhance survival, proliferation, and resistance to therapy. By reshaping these pathways, malignant cells can escape apoptosis, fuel metabolic adaptations, restore redox balance, and even promote immune evasion (Kreß et al, 2023). Thus, dramatic advances in understanding the implications of mitochondrial metabolism in cancer, have set the focus on intervention targeting mitochondria as a therapeutic strategy in oncology (Fulda et al, 2010; Modica-Napolitano and Singh, 2004; Sainero-Alcolado et al, 2022).

Intriguingly, in the context of cancer, mitohormesis represents a double-edged sword. Upregulation of mitochondrial stress responses could serve as an effective cytoprotective mechanism that cancer cells can adopt to adapt and promote tumor progression, drug resistance, and invasiveness. Elevated UPR^mt activation in a subset of breast cancer patients has been shown to result in significantly worse survival and metastatic features (Kenny et al, 2019). A recent comprehensive review also tackles the pressing question of whether the complex processes of UPR^mt could have crossover signaling pathways with cell death mechanisms, emphasizing that the magnitude and the tolerance of the stress experienced by cells could determine their fate and outcome in the cancer setting (Zhang et al, 2024). While some recent studies have highlighted that ISR activation promotes tumorigenesis (Ghaddar et al, 2021; Verginadis et al, 2022) or cancer progression (Cerqua et al, 2025), the specific stress kinase that triggers the ISR in cancer cells can differentially affect the outcome of ISR activation. PKR activation in cancer is context-dependent: in some tumor cells, PKR promotes tumor cell survival and malignant transformation (Donzé et al, 1995), while in others it acts as a tumor suppressor, inducing cell cycle arrest or cell death, slowing tumor growth (Darini et al, 2019; Yoon et al, 2009; Zamanian-Daryoush et al,

1999). These differential outcomes depend on the cancer cell type, the initial stressor that activates PKR, and the downstream players or pathways engaged (Koromilas, 2015). On the other hand, PERK activation, along with induction of ER stress, has been strongly associated with induction of tumorigenesis (Bi et al, 2005). These complex nuances must be considered for different cancer types in order to target ISR and UPR^mt in cancer therapeutics.

GDF15, earlier described for its beneficial effects in obesity-related disorders, plays a paradoxical role in cancer. In the context of cancer, GDF15 takes on a more detrimental role, particularly in cancer-associated cachexia. Research has demonstrated that elevated levels of GDF15 are upregulated in specific cancer subsets, contributing to systemic energy imbalance, muscle wasting and related weight loss (Siddiqui et al, 2022). In a recent clinical study involving patients with cancer cachexia and high GDF15 levels, inhibiting GDF15 using a humanized monoclonal antibody against GDF15 led to notable improvements in weight, appetite, and physical activity, while also lowering serum GDF15 levels.(Groarke et al, 2024). This contrast in GDF15 function highlights its disease context-specific, dependent effects, beneficial in obesity treatment but potentially harmful in defined cancer.

Interestingly, the NAT analog, N-acetyloxfenicine (NAO), has recently been explored as a drug that can induce mitohormesis, causing inhibition of tumor growth. NAT triggers mild mitochondrial ROS (mtROS) formation, causing activation of the ISR ultimately leading to the upregulation of antioxidant genes resulting in heat tolerance and inhibition of tumor growth when supplemented exogenously (Matsumura et al, 2020), as discussed in the previous section. To utilize the beneficial properties of NAT for treatment of cancer, the same group generated a synthetic functional analog of NAT, NAO (Matsumura et al, 2023). Pretreatment with NAO reduced the growth of human colorectal tumor (HCT) cells implanted in mice. These findings suggest that NAO may be a potential therapeutic agent due to its antioxidant properties and ability to inhibit tumor growth.

While these findings are promising, it is crucial to carefully consider the inhibition or activation of mitohormesis in the context of the specific pathophysiological and biomolecular characteristics of different cancers. To develop tailored mitohormesis-based therapies in oncology, it is crucial to decipher the molecular signatures of disparate cancer types and patient-specific variations, as well as individual drug responses. The rapid advancement of omics technologies is revolutionizing cancer characterization, moving beyond broad classifications of cancer subtypes to precise, patient-specific molecular landscapes. Integrating genomics, transcriptomics, proteomics, and metabolomics will allow researchers to uncover ISR vulnerabilities unique to each tumor and identify potential biomarkers for therapeutic intervention.

## Neurodegenerative and cognitive disease

Mitochondrial dysfunction and mitochondrial stress, especially oxidative, are hallmarks of many neurodegenerative diseases such as Alzheimer's, Parkinson's and Huntington's disease (Lin and Beal, 2006). Interestingly, antioxidants have not yielded significant clinical benefits for treatment of neurodegenerative disease (Morén et al, 2022), likely due to poor bioavailability, inconsistent dosing, or unclear uptake and targeted pathways. As a result, alternative therapeutic approaches, such as those aimed at inducing

mitohormesis to alleviate mitochondrial stress and enhance brain function, represent promising targets for neurodegenerative disease treatment.

Altering mitochondrial ATP and ROS production is one way mitohormesis inducers could induce mitochondrial stress signaling to improve brain function. Scott et al identified N-propargylglycine (N-PPG) as an irreversible inhibitor of proline dehydrogenase (PRODH), an important flavin-dependent oxidoreductase involved in proline catabolism to glutamate, that promotes transfer of electrons to the ETC to produce ATP or ROS (Scott et al, 2019). PRODH is also a p53-inducible flavoprotein that is relevant for cancer cell metabolic adaptation under stress conditions. Interestingly, in the initial study, inhibition of PRODH activity through N-PPG was able to induce mitochondrial proteostasis and upregulate UPRᵐᵗ-associated genes in breast cancer models, independent of its anticancer activity. Given these promising findings in cancer cells, the authors hypothesized that N-PPG could be used as a mitohormesis inducer in the brain (Scott et al, 2021). In wild-type mice, N-PPG was able to cross the blood-brain barrier (BBB) and induce UPRᵐᵗ-associated gene expression. Additionally, RNA-seq Gene Ontology (GO) analysis of brains from treated mice revealed upregulation of pathways related to stimulated neural cell function. N-PPG treatment in a Huntington's disease mouse model allowed for the Huntington disease brain transcriptome to be normalized to resemble WT brains more closely (Teramayi et al, 2024). This included increases in expression of genes that are normally dysregulated in Huntington's disease, such as tyrosine hydroxylase (*Th*), dopamine receptor-1 (*Drd1*) and adenosine A2A receptor (*Adora2a*).

Another approach to targeting brain mitochondria for neuroprotection, involves using a caloric restriction (CR) mimetic. CR has been shown to cause neuroprotection in primates with Parkinson's disease (Maswood et al, 2004). However, since CR is difficult to maintain in humans, CR mimetics have been explored to mimic the metabolic and physiological effects of CR without decreasing calorie intake. 3-bromopyruvate (3-BP), an inhibitor of hexokinase, key enzyme in glycolysis, has been explored for its ability to improve brain function in aged rats (Arya et al, 2022). 3-BP treatment, in both young and aged male rats, increased ROS production in brain tissue due to upregulation of mitochondrial ETC complex I and IV activity. However, due to this mild induction of mtROS production, antioxidant activity was noted to be increased with 3-BP treatment. The authors also found an upregulation of autophagy markers associated with 3-BP's ability to inhibit acetyltransferase. Autophagy is associated with hormesis because clearing of damaged organelles in neurons results in protection from ROS-induced damage, which is essential for brain function (Hara et al, 2006; Komatsu et al, 2006). Moreover, neuron-specific enolase (NSE), a neuronal marker for structural damage, was decreased in aged rat brain, but its expression was restored with 3-BP treatment. Overall, mild mtROS induction in this model, triggered adaptive antioxidant and autophagy pathways that benefited aged rat brains, showing the promise of targeting mitohormetic pathways for neuroprotection.

While activation of the ISR/UPRᵐᵗ has recently been explored as a treatment avenue for neurodegenerative disease, other studies have also shown that inhibiting the ISR via the small molecule ISRIB can also be neuroprotective. ISRIB inhibits the ISR via binding to eIF2α's guanine nucleotide exchange factor (GEF)

eIF2B. This binding allosterically antagonizes the inhibitory effect of phosphorylated eIF2α, preventing activation of the ISR (Zyryanova et al, 2021). ISRIB was first found in a screen seeking to identify PERK inhibitors, and the authors found that ISRIB treatment in wild-type mice resulted in improved spatial learning and fear response (Sidrauski et al, 2013). Further, in mice with a traumatic brain injury, ISRIB treatment improved cognitive function even four months following injury and treatment (Chou et al, 2017). ISRIB and other ISR inhibitors have recently entered human clinical trials for the treatment of neurodegenerative disease.

## Lifespan and healthspan

Aging-related molecular mechanisms contribute to a wide range of diseases. Uncovering these mechanisms, which often involve changes to mitochondrial functions, undoubtedly represents an attractive field. Pioneering studies on invertebrate models such as *C. elegans* (Durieux et al, 2011; Mao et al, 2019; Shao et al, 2016) have provided valuable insights into the effects of mitohormesis on increasing lifespan. Although the first mammalian model of mitohormesis demonstrated that mitochondrial stress could extend lifespan in mice (Lapointe et al, 2009), most mammalian studies of mitohormesis focus on extending healthspan or improving age-related diseases. The role of mitohormesis in this context remains a topic of debate. For example, it remains largely unclear whether the induction of mitochondrial stress contributes to or alleviates aging. Mild mitochondrial stress signaling may offer protective effects; however, excessive stress signaling can result in detrimental effects (Burtscher et al, 2023).

Recently, several studies revolved around these puzzling questions. A recent study focused on identifying compounds that can trigger transient and reversible mitochondrial stress, to extend lifespan in invertebrates and improve aging phenotypes in mice (Costa-Machado et al, 2023). The authors employed an optimized in vitro screening platform, ultimately identifying compounds able to induce transient, sublethal mitochondrial stress, eliciting improved mitochondrial function. Through this process, the researchers identified harmol, which showed the most potent effects and was subsequently selected for further characterization in vivo. Chronic treatment for three months in a DIO mouse model showed reduced body weight and fat mass, and improved glucose homeostasis. Moreover, harmol-treated mice had decreased accumulation of lipid content in the liver and increased metabolic fitness. Harmol treatment extended lifespan in two independent invertebrate models, *C. elegans* and *Drosophila melanogaster*, and treatment of old mice ameliorated frailty development. The authors found that harmol functions through an alteration in the gamma-aminobutyric acid (GABA) signaling and by exerting an inhibitory effect on monoamine oxidase-B (MAO-B) to induce mitohormesis. This suggests that harmol shares molecular targets with antidepressant stimuli, such as drugs, exercise, or CR, indicating a possible link between psychological well-being, mitochondrial function, and healthspan. The role of mitohormesis in the context of mouse lifespan was also leveraged in an overexpression model of the mitochondrial transcription factor A (TFAM) (Igami et al, 2024). The authors explored different models of TFAM overexpression (WT, homozygous (Tg/Tg), and heterozygous (Tg)). TFAM overexpression, especially in the homozygous (TFAM Tg/

Tg) model, altered gene and protein expression in mitochondrial ETC complexes, ultimately leading to increased levels of ATF4, FGF21 and GDF15, resulting in leaner, smaller mice with extended lifespan.

# Concluding remarks

Recent insights into mitochondrial functions and pathways downstream of mitochondrial stress have uncovered the importance of these organelles as a signaling hub. These findings shed light on how mitochondria can not only communicate intracellularly between different organelles, but also participate in paracrine and endocrine signaling (Chandel, 2014; Jiang et al, 1999). This culminates in distal effects and responses that ultimately exert organismal benefits. This signaling and communication capacity of mitochondria is at the core of mitohormetic pathways. Although several studies have been carried out in invertebrate organisms, only recently has progress been made to study mitohormesis in mammalian organisms. Herein, we attempt to summarize the most recent findings on the beneficial effects of mitohormesis in mammalian models and their possible translational applications. Indeed, we highlighted how these pathways can eventually be harnessed for therapeutic use in the treatment of a plethora of diseases associated with mitochondrial stress or dysfunctions.

As introduced earlier, a mild or transient stressor affecting mitochondrial function is at the core of the signaling cascade that results in the induction of ISR and UPRmt. Rather than inducing complete organismal dysfunction, this mild or transient stressor can be resolved by stress response pathways. The induction of the ISR and UPRmt ultimately leads to the upregulation of genes that participate in stress resolution (Münch, 2018; Zhao, 2002). This stress resolution pathway varies based on the type of mitochondrial stressor, the affected tissue type and the degree of stress. In some instances, the stress can be resolved locally. In the models discussed above, an antioxidant response is often critical to stress resolution (Ahola et al, 2022; Cox et al, 2018; Lee et al, 2023), involving genes such as *Nrf2*, as well as those involved in GSH metabolism or mitochondrial antioxidant enzymes, such as SOD2. The anti-inflammatory pathway can also be indirectly upregulated due to UPRmt or ISR induction, allowing for stress resolution. However, unlike the antioxidant response, these anti-inflammatory pathways do not have a consistent set of upregulated genes. Timblin et al described how UPRmt induction caused changes in acetyl-CoA levels, leading to altered histone acetylation of inflammatory genes. While Zhang et al noted that UPRmt induced changes in mitochondria to ER contact sites resulting in decreased activation of UPRER and less inflammation. Meanwhile, mitochondrial signaling can result in systemic changes usually involving a secreted factor or effector that causes changes to whole-body metabolism. Well-characterized mitokines, such as FGF21 and GDF15, are known for their significant role in enhancing systemic metabolism Although they exhibit some overlaps in their effects, they still have distinct signatures (Kang et al, 2021b). On the other hand, less characterized mitokines, like MOTS-c, have been linked to the beneficial effects of exercise (Lee et al, 2015; Reynolds et al, 2021), while NAT has been shown to upregulate antioxidant pathways through the FoxO–Keap1 signaling axis, offering protection from heat or restraint stress (Matsumura et al, 2020). A few studies have described interorgan mitohormesis, where mitochondrial stress in one tissue leads to the

secretion of signaling factors that trigger compensatory responses in distal organs. Unique signaling factors have been described in these studies. Crewe and colleagues, for instance, observed secretion of mitochondrial-derived sEVs from adipose tissue experiencing iron deficiency, which ultimately led to heart protection against IR injury. Furthermore, studies by Chang et al and Kang et al highlighted that iWAT represents a critical compensatory tissue in mammalian mitohormesis. iWAT browning was enhanced via FGF21 secretion from BAT in the Chang et al study. Meanwhile, the study by Kang et al, revealed that the exercise mimetics MOTS-c and β-END, secreted from POMC neurons, promoted browning via increased sympathetic innervation into iWAT.

Targeting mitohormesis as a therapeutic approach is still in its early stages. While many research groups are actively developing drugs to trigger mitohormesis and are studying their effects in preclinical models, the underlying mechanisms remain not fully understood. As a result, there is currently only one clinical trial exploring the use of magnetic-field therapy to activate mitohormesis for treating T2DM (Franco-Obregón et al, 2023). In contrast, numerous clinical trials have investigated antioxidant therapies for diseases such as atherosclerosis, neurodegenerative disorders, and cancer. Most of them have proven to have unsuccessful or unclear outcomes (Ladas et al, 2004; Morén et al, 2022; Steinhubl, 2008). These inconsistencies could be related to multiple factors such as dosing, formulation, or timing of administration. More importantly, the limited efficacy of antioxidant therapy is mostly due to their complex and yet poorly defined mechanisms of action, which often involve multiple biological processes and targets, including potential interference with mitohormetic pathways, an area still largely unexplored, especially for antioxidants. In addition, the effects of antioxidants appear to be context-dependent and, in some cases, even controversial. For instance, in certain circumstances such as cancer, antioxidants may exhibit pro-oxidative effects, while in other conditions, they might suppress beneficial adaptive responses to oxidative stress (Hecht et al, 2024; Poljsak et al, 2013; Ristow and Schmeisser, 2014). In fact, it remains unclear whether, in specific diseases, it is more advantageous to enhance antioxidant defenses or allow sublethal controlled oxidative stress to stimulate protective pathways such as mitohormesis. Some studies suggest that mild ROS-inducing therapies may offer a promising alternative for disease treatment (Forman and Zhang, 2021; Ristow et al, 2009). These discrepancies underscore a critical need to advance the field and clarify these questions. Continued investigation using preclinical models, followed by human clinical trials, is crucial to identify which specific diseases might benefit from mitohormetic therapies. To address these pressing challenges, omics data, integrated with genome-wide association studies (GWAS) and published studies, could be leveraged to understand if mitohormesis is tied to specific disease states and could be targeted in the development and advancement of pathologies. This highlights a move toward personalized medicine, where mitohormesis mechanisms are leveraged to tailor therapeutic strategies based on individual disease profiles, dramatically advancing how we treat multiple pathological conditions associated with mitochondrial dysfunction.

Since the concept of "mitohormesis" was introduced in 2006 (Tapia, 2006) and the first study on it in mammals appeared in 2009 (Lapointe et al, 2009; Ristow et al, 2009), the field has seen significant growth. As research dynamically translates from invertebrate and cell-

based models to exploring mitohormetic mechanisms in complex mammalian organisms, it is essential to bridge in vitro and in vivo studies, recognizing the complementary perspectives of both approaches. This integration is necessary to fully understand the underlying mechanisms and broader effects of mitohormesis. Regardless of these advancements, there are still considerable gaps in our understanding of the specific mechanisms that shape the response from organelles to the whole organism in response to mitochondrial stress. Thus far, studies in mammalian models have mainly focused on the liver, adipose, and muscle. A few studies have addressed other tissue types, such as immune cells, neuronal subtypes, skeletal muscle, and the heart. In addition, our literature search identified specific mechanistic patterns associated with mitochondrial stress responses at both the local tissue level and whole-body level across the variety of papers discussed. To clarify whether these represent conserved mitohormetic pathways, further investigation is necessary to understand why specific mitochondrial stressors evoke different responses. Besides, identifying the particular mitokines secreted from stressed tissue and what specific pathways they upregulate to influence systemic metabolism and compensatory pathways in peripheral tissues, poses several challenges. There is an urgent need for more comprehensive screening to discover novel mitokines, as well as for developing tools to analyze the secretomes of specific tissue types, especially in the context of disease states. Understanding the distinct roles of the UPR$^{mt}$ and ISR$^{mt}$ adds another layer of complexity to mitohormetic models in mammals. Expanded research is essential to unravel how the induction of mitochondrial stress activates various ISR and UPR mechanisms, especially in comparison to other forms of cellular stress.

One of the biggest roadblocks in drug development is identifying molecules that selectively target a specific pathway. The drug discovery research in mitohormesis clearly exemplified these complexities, as many of the current models employ stressors that can also provoke stress responses in organelles beyond mitochondria and activate multiple stress response mechanisms simultaneously. Thus, it is critical to create biochemical tools that pinpoint key pathways unique to stress originating from individual organelles and identify precise molecular targets. A recent study addresses this crucial issue using a synthetic dimerizable PERK construct to selectively activate ISR in a tunable way, while avoiding the simultaneous activation of a parallel pathway. This approach helped to define specific ISR responses, revealing that ATF4 plays a central role in rerouting biosynthetic pathways to enable amino acid synthesis during PERK-induced ISR activation (Labbé et al, 2024). On the same note, a study from Gradjean et al introduced a transcriptomic "fingerprint" approach for stress responses. The author developed a medium-throughput target RNA-seq assay designed for reporting the activation of major stress-responsive proteostasis pathways, including ISR and UPR, induced by various genetic and pharmacological agents. This study provided a powerful tool for identifying compounds with high specificity earlier in the screening pipeline, distinguishing between different stress responses and their corresponding gene expression signatures (Grandjean et al, 2019).

In summary, the studies presented in this bird's-eye view of the latest research on mammalian mitohormesis clearly demonstrate that integrating findings in vitro and in vivo is advancing this exciting field. These advances are bringing mitohormesis mechanisms to light with high translational potential that could profoundly alter the landscape for treating disease states often linked to mitochondrial stress in humans.

# Peer review information

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

## Acknowledgements

This work is supported by NIH grant 1R01DK126830-01 to AS. ALG was supported by the American Heart Association predoctoral fellowship 23PRE1018992 (https://doi.org/10.58275/AHA.23PRE1018992.pc.gr.161143). Due to the limited space available for this review and the vast amount of research on the subjects covered, it might be possible that some important studies were not included. The authors would like to express their gratitude to all the researchers whose work has played a significant role in advancing this field and apologize for any unintentional omissions.

## Author contributions

**Amanda L Gunawan**: Conceptualization; Visualization; Writing—original draft; Writing—review and editing. **Irene Liparulo**: Conceptualization; Visualization; Writing—original draft; Writing—review and editing; Author ALG and Author IL contributed equally. **Andreas Stahl**: Conceptualization; Funding acquisition; Visualization; Writing—original draft; Writing—review and editing.

## Disclosure and competing interests statement

The authors declare no competing interests.

