## [Peer Review File · The EMBO Journal]

Mammalian mitohormesis: from mitochondrial stressors to organismal benefits

Amanda Gunawan, Irene Liparulo, and Andreas Stahl

Corresponding authors: Andreas Stahl (astahl@berkeley.edu) , Irene Liparulo (irene.liparulo@berkeley.edu)

Review Timeline:

Submission Date:	8th Nov 24
Editorial Decision:	16th Dec 24
Revision Received:	14th Apr 25
Editorial Decision:	2nd Jul 25
Revision Received:	11th Jul 25
Accepted:	25th Jul 25

Editor: Daniel Klimmeck

Transaction Report:

Dear Andreas,

Thank you again for sending us your review article for consideration, as well as for your patience with our feedback at this time.

We have now received dedicated input on your manuscript from two dedicated experts. As you will see, they appreciate the review and find it timely and worth publishing. They also provide constructive feedback on how to further improve and amend it by advancing the introduction and discussion, as well as clarifying a number of aspects. Further, they give advice on how to enhance overall structure of the piece, and complement the literature context cited.

I hope you will find the comments helpful. I am sure that an amended version incorporating the suggestions made by the referees will be highly noted and appreciated. I would thus like to invite you to submit such a revised version using the link enclosed below.

Please let me know in case I can be of any help with this.

with
Best wishes to Berkeley,
Daniel

Daniel Klimmeck, PhD
Senior Editor
The EMBO Journal

Referee #1:

Review Referee Report
EMBOJ-2024-119575, corr. author Prof. Stahl
"Mammalian mitohormesis and the unexpected path from mitochondrial stressors to organismal benefits"

The review manuscript from Gunawan et al. aims to summarise the research on Mitohormesis and mitochondrial stress responses. The authors describe Integrated stress response and mtUPR in different mammalian models and suggest that targeting these pathways could have therapeutic potential. However, the overall message of the review remains rather unclear to the reader due to the writing's somewhat confusing structure and flow. I suggest major revision for the manuscript and would not at this stage even go into too many detailed comments before some rewriting and structural changes to the paper. The manuscript also needs major editing of the general writing (examples below).

Major comments

Overall, the manuscript is difficult to read and follow. It is, in my opinion, too long and poorly structured and the writing is generalized without clear scientific statements or conclusions. Citations are often placed misleadingly (or completely in the wrong places) and the authors have forgotten several key papers from the field that have majorly contributed to our understanding of mammalian ISR. A major issue comes already in the beginning with explaining the main players of the manuscript. It is unclear what the authors mean by ISR and UPR and how these two differ from each other in mammalian system. This is of great importance as mammalian mtUPR is not very well defined and often thought that it doesn't properly exist without ISR. This should be the structure and the main message of the Introduction section. The writing currently remains as a list of papers that have mentioned ISR or mtUPR in the past.

Minor comments

- Is it necessary to introduce "mitochondrial information processing system (MIPS)" "mitochondrial precursor overaccumulation

stress (mPOS)" "compromised protein import response (mitoCPR)", "mitochondrial-specific stress response (MSR)" etc. when they are not mentioned again in the manuscript and are not widely used or accepted terms in the field?

- "Recently, the concept of the mitochondrial information processing system (MIPS) introduced a novel perspective of mitochondria as communicating organelles." And then later authors state that 1996 mitochondria were shown to communicate and participate in cell death decisions. I would say they idea is not very novel any more
 - "Most of the studies were initially conducted in *C. Elegans* since the genetic manipulation of the folding environment in the mitochondria in these organisms was easier to achieve(Haynes et al, 2007)." I don't think this paper from Haynes states that they use *C.Elegans* for the reason that it is easy.
 - "Attenuation of global protein synthesis by activation of eukaryotic translation initiation factor 2 (eIF2) complex subunit 2 α (eIF2 α)(Anderson & Haynes, 2020; Forsstrom et al, 2019; Ost et al, 2020) also plays a key role in this cascade." What key role? Please introduce the eIF2a cascade. Also not initially discovered by these papers referred to here.
 - "These triggers caused low-grade mitochondrial stress which the tissue type can resolve locally via activation of cellular stress pathways." If the authors refer to Cox10^{-/-}, it is not low-grade stress. Mice die at the age of 4weeks due to complete loss of CIV function
 - "In Ahola et al. the authors describe perturbations to the electron transport chain (ETC) via knockout of Cox10 (complex IV) in the muscle and heart of mice (Ahola et al, 2022) which ultimately leads to an antioxidant response that protects the heart from ferroptosis, a form of cell death. Although the mitochondria displayed dysfunctions including changes to mitochondrial morphology and decreased COX activity, the ISR was shown to be activated via the Oma1-DELE1-ATF4 axis(Guo et al, 2020). This pathway is a key mechanism that allows mitochondrial stress triggers to activate the ISR in mammalian cells. Activation of the ISR induced expression of trans-sulfuration sulphuration pathway enzymes able to increase selenium levels within MEFs and a higher GSH/GSSG ratio. This resulted in increased levels of Gpx4 a protein that utilizes GSH to reduce lipid peroxide radicals to prevent ferroptosis(Ursini & Maiorino, 2020)." Here is a good example how the references are misplaced. Ahola et al has shown Cox10-Oma1-Dele1-ATF4-GPX4 pathway. Guo et al (together with Fessler et al 2020) were the first to identify Oma1-Dele1-ATF4, Ursini&Maiorino is a review where they did not discover the link between Gpx4 and ferroptosis.
 - "Interestingly, mitochondrial stress in the liver is a biomarker for metabolic dysfunction-associated steatotic liver disease (MASLD) in patients." Is mitochondrial stress a biomarker? Or some metabolic outcome of this. This sentence is just wrong.
 - "In this study by Pereira et al, FGF21 is identified as the mitokine causing systemic metabolic changes, however rather than describing one particular target tissue, they hypothesize that multiple peripheral organs could contribute to the overall mitohormesis effects observed." FGF21 was identified as mitokine by Suomalainen group and later Trifunovic.
- Several other examples like this. Please, carefully check the facts and the writing.

Sentences that need editing. Listing some examples from the text:

- Intriguingly, it has also been uncovered the ability of mitochondria to specialize in subpopulations to meet and maintain specific bioenergetic needs within a given cell
- When the stress overwhelms the adaptive response and fixing the damages evoked by the insults could be more costly in energy or less beneficial, cells could adopt other strategies leading to clearance of the organelle (mitophagy) or cell death mechanisms such as apoptosis.
- The potential to treat and prevent various diseases, from metabolic disorders to aging, through harnessing mitohormetic mechanisms is promising.
- "An alternative pathway whereby stress responses allow for local beneficial phenotypes is via upregulation of anti-inflammatory pathways. In another paper the authors show that mitochondrial stress responses can protect from the low-grade inflammation usually associated with obese states."
- Several other examples! Please re-write!

Referee #2:

This review article it a well-balanced and concise overview about the concept of mitohormesis particularly in mammals, i.e. excluding earlier evidence from invertebrate model organisms.

I only have a few minor suggestions all of which should be considered optional:

The idea of a non-linear dose-response interaction in biological systems, and namely humans (i.e. mammals) dates back to Paracelsus, who in my opinion deserves a honourable mention in the introduction.

Similarly, the predecessor of the term mitohormesis, namely hormesis, was mainly reinstated by Ed Calabrese, who also should be cited, also given the multiple excellent reviews he has published, including the pertinent PubMedID 12610596.

Lastly and regarding Table 1, exercise as a mitochondrial stressor has been linked to mitohormesis in humans in 2009 & shown to exert oxidative stress response as well as improved glucose metabolism. The corresponding publication (PubMedID 19433800) might be listed as such in Table 1.

Referee #1:

Review Referee Report

EMBOJ-2024-119575, corr. author Prof. Stahl

"Mammalian mitohormesis and the unexpected path from mitochondrial stressors to organismal benefits"

The review manuscript from Gunawan et al. aims to summarise the research on Mitohormesis and mitochondrial stress responses. The authors describe Integrated stress response and mtUPR in different mammalian models and suggest that targeting these pathways could have therapeutic potential. However, the overall message of the review remains rather unclear to the reader due to the writing's somewhat confusing structure and flow. I suggest major revision for the manuscript and would not at this stage even go into too many detailed comments before some rewriting and structural changes to the paper. The manuscript also needs major editing of the general writing (examples below).

Major comments

Overall, the manuscript is difficult to read and follow. It is, in my opinion, too long and poorly structured and the writing is generalized without clear scientific statements or conclusions. Citations are often placed misleadingly (or completely in the wrong places) and the authors have forgotten several key papers from the field that have majorly contributed to our understanding of mammalian ISR. A major issue comes already in the beginning with explaining the main players of the manuscript. It is unclear what the authors mean by ISR and UPR and how these two differ from each other in mammalian system. This is of great importance as mammalian mtUPR is not very well defined and often thought that it doesn't properly exist without ISR. This should be the structure and the main message of the Introduction section. The writing currently remains as a list of papers that have mentioned ISR or mtUPR in the past.

We sincerely appreciated the reviewer's valuable feedback and suggestions aimed at enhancing the review's structure and clarifying its objectives. In response, we addressed the main concerns by clearly distinguishing ISR and UPR, outlining their roles in mammalian systems, and providing a brief overview of current knowledge. We made a dedicated section to uncover these aspects (see new section: *Coordinated stress responses: understanding ISR and UPR^{mt} mechanisms in mitochondrial adaptation*). To summarize these pathways, we also included a graphical abstract highlighting the pathway cascades and key players (**Fig.3**). In this amended version, we made significant revisions to clarify previously unclear passages and eliminated any unnecessary redundancies throughout the text.

Minor comments

- Is it necessary to introduce "mitochondrial information processing system (MIPS)" "mitochondrial precursor overaccumulation stress (mPOS)" "compromised protein import response (mitoCPR)", "mitochondrial-specific stress response (MSR)" etc. when

they are not mentioned again in the manuscript and are not widely used or accepted terms in the field?

We introduced these terms to provide a broad overview of the mechanisms involved and how recent research has reshaped our understanding of mitochondria's role in health and disease settings. However, we recognized that these terms are not widely accepted, and we removed part of this section from the manuscript.

- "Recently, the concept of the mitochondrial information processing system (MIPS) introduced a novel perspective of mitochondria as communicating organelles." And then later authors state that 1996 mitochondria were shown to communicate and participate in cell death decisions. I would say they idea is not very novel any more

We have revised this section to highlight that, although the term "MIPS" was coined recently, the idea of mitochondria as a communicating organelle is not novel.

- "Most of the studies were initially conducted in *C. Elegans* since the genetic manipulation of the folding environment in the mitochondria in these organisms was easier to achieve(Haynes et al, 2007)." I don't think this paper from Haynes states that they use *C.Elegans* for the reason that it is easy.

We apologize for any confusion our previous statements may have caused. It was not our intention to suggest that genetic manipulation of these organisms is inherently simpler. As the authors of the paper referred (Haynes et al, 2007) pointed out, "We have chosen to study the UPRmt in *C. elegans* because of the ease with which the response can be elicited by genetic manipulation of the folding environment in the mitochondria," as noted by Yoneda et al. (2004). We will revise this sentence to clarify that, given the tools and methods available nearly two decades ago, alongside existing research, manipulating genes in *C. elegans* was relatively more straightforward. The authors took advantage of the straightforward nature of gene function manipulation in *C. elegans* compared to other organisms.

- "Attenuation of global protein synthesis by activation of eukaryotic translation initiation factor 2 (eIF2) complex subunit 2 α (eIF2 α)(Anderson & Haynes, 2020; Forsstrom et al, 2019; Ost et al, 2020) also plays a key role in this cascade." What key role? Please introduce the eIF2a cascade. Also not initially discovered by these papers referred to here.

We clarified the key role of eIF2 α in the ISR cascade, and we have updated the references.

- "These triggers caused low-grade mitochondrial stress, which the tissue type can resolve locally via activation of cellular stress pathways." If the authors refer to Cox10^{-/-}, it is not low-grade stress. Mice die at the age of 4weeks due to complete loss of CIV function

We thank the reviewer for pointing out this oversight, and we updated the text accordingly. We highlighted that Cox10^{-/-} mouse model, as described in Ahola et al. (2022), exhibits profound mitochondrial dysfunction due to the complete loss of complex IV activity. This results in severe cardiomyopathy and early mortality. Thus, the mitochondrial stress in this context is clearly high-grade rather than low-grade.

However, the Oma1-mediated ISR in this model exerts cardioprotective effects by mitigating ferroptosis.

• "In Ahola et al. the authors describe perturbations to the electron transport chain (ETC) via knockout of Cox10 (complex IV) in the muscle and heart of mice (Ahola et al, 2022) which ultimately leads to an antioxidant response that protects the heart from ferroptosis, a form of cell death. Although the mitochondria displayed dysfunctions including changes to mitochondrial morphology and decreased COX activity, the ISR was shown to be activated via the Oma1-DELE1-ATF4 axis(Guo et al, 2020). This pathway is a key mechanism that allows mitochondrial stress triggers to activate the ISR in mammalian cells. Activation of the ISR induced expression of trans-sulfuration sulphuration pathway enzymes able to increase selenium levels within MEFs and a higher GSH/GSSG ratio. This resulted in increased levels of Gpx4 a protein that utilizes GSH to reduce lipid peroxide radicals to prevent ferroptosis(Ursini & Maiorino, 2020)." Here is a good example how the references are misplaced. Ahola et al has shown Cox10-Oma1-Dele1-ATF4-GPX4 pathway. Guo et al (together with Fessler et al 2020) were the first to identify Oma1-Dele1-ATF4, Ursini&Maiorino is a review where they did not discover the link between Gpx4 and ferroptosis

To address the issue raised by the reviewer, we fixed the misplaced references.

• "Interestingly, mitochondrial stress in the liver is a biomarker for metabolic dysfunction-associated steatotic liver disease (MASLD) in patients." Is mitochondrial stress a biomarker? Or some metabolic outcome of this. This sentence is just wrong. We apologize for the misunderstanding and incorrect phrasing. We have updated the text to emphasize the possible etiologies of MASLD related to mitochondria.

• "In this study by Pereira et al, FGF21 is identified as the mitokine causing systemic metabolic changes, however rather than describing one particular target tissue, they hypothesize that multiple peripheral organs could contribute to the overall mitohormesis effects observed." FGF21 was identified as mitokine by Suomalainen group and later Trifunovic.

We outlined the timelines for key milestones in the discovery and characterization of FGF21.

• Several other examples like this. Please, carefully check the facts and the writing. We thoroughly reviewed the manuscript and clarified any ambiguous sections.

Sentences that need editing. Listing some examples from the text:

- Intriguingly, it has also been uncovered the ability of mitochondria to specialize in subpopulations to meet and maintain specific bioenergetic needs within a given cell
- When the stress overwhelms the adaptive response and fixing the damages evoked by the insults could be more costly in energy or less beneficial, cells could adopt other strategies leading to clearance of the organelle (mitophagy) or cell death mechanisms such as apoptosis.
- The potential to treat and prevent various diseases, from metabolic disorders to aging,

through harnessing mitohormetic mechanisms is promising.

- "An alternative pathway whereby stress responses allow for local beneficial phenotypes is via upregulation of anti-inflammatory pathways. In another paper the authors show that mitochondrial stress responses can protect from the low-grade inflammation usually associated with obese states."

- Several other examples! Please re-write!

We appreciated the review for highlighting these unclear sentences and brought to our attention potential misunderstandings. We have thoroughly revised and rephrased various passages throughout the text to improve clarity and enhance overall readability.

Referee #2:

This review article is a well-balanced and concise overview about the concept of mitohormesis particularly in mammals, i.e. excluding earlier evidence from invertebrate model organisms.

I only have a few minor suggestions all of which should be considered optional:

The idea of a non-linear dose-response interaction in biological systems, and namely humans (i.e. mammals) dates back to Paracelsus, who in my opinion deserves a honourable mention in the introduction.

We truly valued the insightful suggestion of adding the historical roots of the hormesis concept. We have now mentioned Paracelsus' role in defining the non-linear dose-response interaction in biological systems.

Similarly, the predecessor of the term mitohormesis, namely hormesis, was mainly reinstated by Ed Calabrese, who also should be cited, also given the multiple excellent reviews he has published, including the pertinent PubMedID 12610596.

We have updated the references accordingly.

Lastly and regarding Table 1, exercise as a mitochondrial stressor has been linked to mitohormesis in humans in 2009 & shown to exert oxidative stress response as well as improved glucose metabolism. The corresponding publication (PubMedID 19433800) might be listed as such in Table 1.

We incorporated the references mentioned per the reviewer's suggestion.

Dear Andreas and colleagues,

Thank you for resubmitting the revised review article manuscript, as well as for your patience with our feedback. As indicated earlier, we have asked both experts to reassess your amended manuscript version. We have received additional comments from referee #1, which I enclose below. Please note that while referee #2 was at this point not able to look back into the review, I have assessed your response to his/her points and found them to be very well addressed.

I am thus pleased to let you know that your review article has been provisionally accepted for publication at the EMBO Journal.

I still need you to consider the additional minor comments of expert #1 regarding writing and citations of previous studies.

We in addition need you to address a number of formatting , annotation points as listed below.

Please note that I have now reached out to our scientific graphics illustrator and his team so they can start preparing the figures for your review. He should contact you on the matter in about two weeks.

I look forward to your feedback on this,
and seeing the piece accepted and at production shortly.

with
Best wishes,
Daniel

Daniel Klimmeck, PhD
Senior Editor
The EMBO Journal

Final formatting adjustments required for this review article manuscript:

- >> Please add up to five keywords to the article.
- >> Add correspondiing author's email address to title page.
- >> Remove the index from the manuscript text.
- >> Add a DISCLOSURE AND COMPETING INTERESTS STATEMENT.

- >> Please upload figure drafts as individual figure files.
- >> Figure legends: remove from the figure files and compile at the end of the manuscript text.
- >> Table 1: will be typeset so please upload an editable version (e.g. .docx).

Referee #1:

The authors have done an excellent job revising the manuscript. The updated version reads more clearly and fluently, with substantial improvements in both structure and scientific clarity as well as the accuracy of the literature references. The reorganization of key sections has greatly enhanced the logical flow of the review, making it more accessible and engaging for readers in the field.

The authors have addressed the previous comments, incorporating recent literature where appropriate and refining the discussion of key concepts. The revised figures and tables are well-integrated and support the narrative effectively.

Overall, the manuscript now provides a comprehensive and insightful overview of the topic and will serve as a valuable resource for researchers in the field. I believe it is suitable for publication after minor revisions to further polish the text.

Minor comments:

Overall, please write out the protein names when you mention them for the first time. For example, Crif1 is not specified anywhere or explained. (line 418)

Line 175 "Short form DELE1 (DELEs) can then be transported out of the mitochondria and into the cytosol through unknown transport mechanisms."

Dele1 is most likely cleaved during import rather than actively transported out (what transporter transports proteins out from mitos?). Please refer to Fessler E, et al. (DELE1 tracks perturbed protein import and processing in human mitochondria. *Nat Commun.* 2022) and Sekine Y et al. A mitochondrial iron-responsive pathway regulated by DELE1. *Mol Cell.* 2023)

Line 227 "Yet, there are other less characterized outputs of the ISR, spanning from mitochondrial peptides, metabolites and small extracellular vesicles (sEVs) which can transport a wide variety of cargo such as miRNAs, mRNA, DNA, proteins, and metabolites and can facilitate mitochondrial transfer (Crewe et al, 2021)" Please reference EV with mitochondrial cargo correctly and then refer to the paper of Crewe et al for mitochondrial EV transfer between tissues (Neuspiel M. et al. Cargo-selected transport from the mitochondria to peroxisomes is mediated by vesicular carriers. *Curr. Biol.* 2008)

Line 268 "In the study by Ahola et al., the authors explore the effects of disrupting the electron transport chain (ETC) by knocking out complex IV subunit Cox10 in the heart and muscle tissues of mice (Ahola et al, 2022)." Please correct "...by knocking out complex IV assembly factor Cox10 in the heart and muscle tissues of.." Cox10 is a heme farnesyltransferase essential for CIV-specific heme, not an actual subunit of COX.

And Line 274 "Activation of the ISR induced expression of trans-sulfuration pathway enzymes, which are able to increase selenium levels within MEFs causing downstream increase in the GSH/GSSG ratio." This is not completely correct. Please rephrase to something in line with: "Activation of the ISR induced expression of trans-sulfuration pathway enzymes, which facilitate glutathione synthesis and the incorporation of selenium into selenoproteins such as Gpx4 within MEFs."

Line 290-296. Please add the reference (Lee et al. 2023) in this part as well.

Lines starting from 385. Could be useful to mention also work of Trifunovic lab investigating the role of FGF21 in DARS2 KO mice (Croon M et al. FGF21 modulates mitochondrial stress response in cardiomyocytes only under mild mitochondrial dysfunction. *Sci Adv.* 2022)

Line 465: "The authors found a mechanism whereby NAT caused production of mitochondrial reactive oxygen species (mROS) by negatively regulating Nrf2 through the FoxO- Keap1 signaling axis. Subsequently, Nrf2 evades Keap1-mediated repression, leading to enhanced expression of antioxidant enzyme genes." This sounds counterintuitive. If NAT increases Keap1 levels, it should inhibit Nrf2 expression and downregulate Nrf2 target genes (and thus increase ROS levels as stated in the first sentence). Please check and correct accordingly.

Line 536 "Although this study was the first paper to fully characterize a mammalian interorgan mitohormesis model, the concept of mitochondrial transfer as a protective mechanism during injury is not a new one." This is rather a bold statement. Isn't systemic FGF21 signalling from myocytes an interorgan mitohormesis as well? Perhaps you can finetune the statement here.

Figure3

Oma1 active site is at the IMS/IM, not matrix substrates. Dele1 has been shown to be processed during the import and then released to the cytoplasm (Jae lab and others). GCN2 can be activated at the lysosome via the amino acid changes such as lower levels of aspartate (Mick et al.)

Table 1

The original Deletor mouse from Suomalainen group is missing from the table. This mouse model was the first to show FGF21 induction in mitochondrial dysfunction. (Tynnismaa et al.)

Referee #1:

The authors have done an excellent job revising the manuscript. The updated version reads more clearly and fluently, with substantial improvements in both structure and scientific clarity as well as the accuracy of the literature references. The reorganization of key sections has greatly enhanced the logical flow of the review, making it more accessible and engaging for readers in the field.

The authors have addressed the previous comments, incorporating recent literature where appropriate and refining the discussion of key concepts. The revised figures and tables are well-integrated and support the narrative effectively.

Overall, the manuscript now provides a comprehensive and insightful overview of the topic and will serve as a valuable resource for researchers in the field. I believe it is suitable for publication after minor revisions to further polish the text.

Minor comments:

Overall, please write out the protein names when you mention them for the first time. For example, Crif1 is not specified anywhere or explained. (line 418)

Protein names are now fully written out when first mentioned in the text, and an abbreviation table has been included for the reader's reference.

Line 175 " Short form DELE1 (DELEs) can then be transported out of the mitochondria and into the cytosol through unknown transport mechanisms."

Dele1 is most likely cleaved during import rather than actively transported out (what transporter transports proteins out from mitos?). Please refer to Fessler E, et al. (DELE1 tracks perturbed protein import and processing in human mitochondria. Nat Commun. 2022 And Sekine Y et al. A mitochondrial iron-responsive pathway regulated by DELE1. Mol Cell. 2023)

We thank the reviewer for their meaningful suggestion regarding the DELE1 cleavage mechanism. The wording in this portion of the review has been updated to denote that OMA1 sits on the inner mitochondrial membrane, where it cleaves DELE1, causing the short form to accumulate in the cytosol. We have also cited the papers that the reviewer suggested, where authors discovered an alternative pathway whereby iron deficiency causes DELE1-mediated activation of HRI independent of OMA1 cleavage. In these studies, iron deficiency causes a DELE1 import block into the mitochondria, leading to full length DELE1 binding to HRI and activation of the ISR^{mt}.

Line 227 "Yet, there are other less characterized outputs of the ISR, spanning from mitochondrial peptides, metabolites and small extracellular vesicles (sEVs) which can transport a wide variety of cargo such as miRNAs, mRNA, DNA, proteins, and metabolites and can facilitate mitochondrial transfer(Crewe et al, 2021)" Please reference EV with mitochondrial cargo correctly and then refer to the paper of Crewe et al for mitochondrial EV transfer between tissues (Neuspiel M. et al. Cargo-selected transport from the mitochondria to peroxisomes is mediated by vesicular carriers. Curr. Biol. 2008)

We have included the wording "mitochondrial derived vesicles" (MDVs) prior to the Crewe et al citation. We have also added the suggested citation Neuspiel et al.

Line 268 "In the study by Ahola et al., the authors explore the effects of disrupting the electron transport chain (ETC) by knocking out complex IV subunit Cox10 in the heart and muscle tissues of mice(Ahola et al, 2022)." Please correct "...by knocking out complex IV assembly factor Cox10 in the heart and muscle tissues of.." Cox10 is a heme farnesyltransferase essential for CIV-specific heme, not an actual subunit of COX.

We apologize for this error and have changed the wording from “subunit” to “assembly factor” for clarity.

And Line 274 "Activation of the ISR induced expression of trans-sulfuration pathway enzymes, which are able to increase selenium levels within MEFs causing downstream increase in the GSH/GSSG ratio." This is not completely correct. Please rephrase to something in line with: "Activation of the ISR induced expression of trans-sulfuration pathway enzymes, which facilitate glutathione synthesis and the incorporation of selenium into selenoproteins such as Gpx4 within MEFs."

The wording in the above line has been revised as suggested.

Line 290-296. Please add the reference (Lee et al. 2023) in this part as well.

We have now added the Lee et al. 2023 reference to this section.

Lines starting from 385. Could be useful to mention also work of Trifunovic lab investigating the role of FGF21 in DARS2 KO mice (Croon M et al. FGF21 modulates mitochondrial stress response in cardiomyocytes only under mild mitochondrial dysfunction. Sci Adv. 2022)

We've added description of the findings related to DARS2 KO mice in this publication in the section describing systemic effects of FGF21. We have also included it in Table 1.

Line 465: "The authors found a mechanism whereby NAT caused production of mitochondrial reactive oxygen species (mROS) by negatively regulating Nrf2 through the FoxO- Keap1 signaling axis. Subsequently, Nrf2 evades Keap1-mediated repression, leading to enhanced expression of antioxidant enzyme genes." This sounds counterintuitive. If NAT increases Keap1 levels, it should inhibit Nrf2 expression and downregulate Nrf2 target genes (and thus increase ROS levels as stated in the first sentence). Please check and correct accordingly.

We have clarified the wording in this section. The authors describe a pathway whereby initially NAT induces mitochondrial ROS production by inhibiting Nrf2 via the FoxO-Keap1 signaling axis, thus inducing mitochondrial stress response activation. However, a second phase of the pathway involves Nrf2 evading inhibition by Keap1, leading to subsequent upregulation of antioxidant genes, which aids in stress tolerance.

Line 536 "Although this study was the first paper to fully characterize a mammalian interorgan mitohormesis model, the concept of mitochondrial transfer as a protective mechanism during injury is not a new one." This is rather a bold statement. Isn't

systemic FGF21 signalling from myocytes an interorgan mitohormesis as well? Perhaps you can finetune the statement here.

We have rephrased the sentences in this section to emphasize the mitochondrial transfer aspect of interorgan mitohormesis rather than making a general statement about mammalian interorgan mitohormesis.

Figure3

Oma1 active site is at the IMS/IM, not matrix substrates. Dele1 has been shown to be processed during the import and then released to the cytoplasm (Jae lab and others). GCN2 can be activated at the lysosome via the amino acid changes such as lower levels of aspartate (Mick et al.)

We highly appreciate the reviewer's suggestion. This figure has been reconfigured to show OMA1 cleaving DELE1 at the IMS/IM leading to short form DELE1 being released into the cytosol. For GCN2 we did not include amino acid deprivation because we are just highlighting mitochondrial activation of the ISR, not general ISR regulation. We have added wording in the text to clarify this starting on line 170.

Table 1

The original Deletor mouse from Suomalainen group is missing from the table. This mouse model was the first to show FGF21 induction in mitochondrial dysfunction. (Tynismaa et al.)

We added the Deletor mouse model to the Figure 1 timeline as an important finding in the mitohormesis field and ensured that this paper is cited in line 464. We recognize that a substantial body of earlier literature has contributed significantly to advancing this field; however, the paper suggested was not included in Table 1 because this table summarizes recent mitohormetic mammalian models from the past 10-15 years. We clarified the specific types of papers listed in Table 1 in the text in lines 310-311. The Deletor mouse does not exhibit a mitohormesis phenotype, and this study was conducted in 2005.

Dear Andreas, dear Irene,

Thank you for submitting the revised version of your review manuscript for consideration by the EMBO Journal.

I have carefully checked your amendments towards the experts' comments and found them to be well addressed and plausibly integrated into the revised text. I am thus very pleased to inform you that your review article has now been accepted for publication in the EMBO Journal.

As mentioned, I have already passed the figures on to our graphics editorial team, who will approach you shortly regarding the versions translated into our journal style.

Your manuscript will be processed for publication by EMBO Press. It will be copy edited and you will receive page proofs prior to publication. Also, you will soon be contacted by Springer Nature to sign your publishing license. When you login to the customer service website, please use the following token to waive the article publication charges: NTGXODIXMTYY. Should you experience any difficulty, please email publishing@embo.org.

If you have any questions, please do not hesitate to contact me or the Editorial Office.

I look forward to progressing swiftly towards online publication of this review article!

with
Best wishes to Berkeley,

Daniel

Daniel Klimmeck, PhD
Senior Editor
The EMBO Journal
EMBO
Postfach 1022-40
Meyerhofstrasse 1
D-69117 Heidelberg
contact@embojournal.org
Submit at: <http://emboj.msubmit.net>